

# Could old tide gauges help estimate past atmospheric variability ?

Paul Platzer[1], Pierre Tandeo[2,3,4], Pierre Ailliot[5], and Bertrand Chapron[1,3]

[1]Laboratoire d'Océanographie Physique et Spatiale, Centre National de la Recherche Scientifique – Ifremer, Plouzané, France
[2]IMT Atlantique, Lab-STICC, UMR CNRS 6285, 29238, Brest, France
[3]Odyssey, Inria/IMT/CNRS, Rennes, France
[4]RIKEN Center for Computational Science, Kobe, 650-0047, Japan
[5]Laboratoire de Mathematiques de Bretagne Atlantique, Univ Brest, UMR CNRS 6205, Brest, France

**Correspondence:** Paul Platzer (paul.platzer@ifremer.fr)

**Abstract.** The storm surge is the non-tidal component of coastal sea-level. It responds to the atmosphere both through the direct effect of atmospheric pressure on the sea-surface, and through Ekman transport induced by wind-stress. Tide gauges have been used to measure the sea-level in coastal cities for centuries, with many records dating back to the 19th-century or even further, at times when direct pressure observations were scarce. Therefore, these old tide gauge records may be used as indirect

observations of sub-seasonal atmospheric variability, complementary to other sensors such as barometers. To investigate this claim, the present work relies on tide gauge records of Brest and Saint-Nazaire, two portal cities in western France, and on the members of NOAA's 20th-century reanalysis (20CRv3) which only assimilates surface pressure observations and uses numerical weather prediction model. Using simple statistical relationships between storm surges and pressure maps, we show that the tide gauge records reveal part of the 19th-century atmospheric variability that was uncaught by the pressure-observations-based

reanalysis. In particular, weighing the 80 reanalysis members based on tide gauge observations indicates that a large number of members are very unlikely, which induces corrections of several tens of Hectopascals in the Bay of Biscay. These findings support the use of early tide gauge records in sensor-scarce areas, both to validate old atmospheric reanalyses and to better probe old atmospheric sub-seasonal variability.

## 1 Introduction

Understanding the atmospheric system requires to understand all scales of variation, from daily to centennial. This cannot be done unless long observation records allow to disentangle these scales. The Twentieth Century Reanalysis Project, hereafter "20CR" (Compo et al., 2011), which is now in its third version, hereafter "20CRv3" (Slivinski et al., 2019), is the only atmospheric reanalysis that runs through the 19th century. It relies on the International Surface Pressure Databank Compo et al. (2019), the largest historical global collection of surface pressure observations, and the NCEP Global Forecast System (GFS)

coupled atmosphere–land model.

    Because it is the longest atmospheric reanalysis available, the 20CR reanalysis is used to study possible long-term trends in atmospheric dynamics (Rodrigues et al., 2018) or for extreme events (Alvarez-Castro et al., 2018). However, although the 20th-century part of 20CR has been compared with other reanalysis (Wohland et al., 2019) and observations (Krueger et al.,



2013), comparisons with independent observations in the 19th century (Brönnimann et al., 2011) are scarce. The present work
is an effort to compare this reanalysis with tide gauge observations. More generally and to our best knowledge, this paper is
the first attempt to use old tide gauges as indirect observations of the atmosphere. However, the opposite direction has been
taken by Tadesse and Wahl (2021), who extended storm surge reconstructions in the past using different atmospheric reanalysis
products, in order to estimate past unobserved extreme storm surges.

Tide gauges are used primarily to measure the tide, which is the largest contributor to sea-level variations in many coastal
cities. The astronomical tide is the result of gravitational attraction of the Sun and Moon on the ocean, combined with Earth's
rotation. It results in periodic rise and fall of the water level (Melchior, 1983), which have been predicted through harmonic
decomposition for centuries. Other physical phenomena impact the water level: a low atmospheric pressure results in a high
sea-level, a phenomenon called the "inverse barometer effect" (Roden and Rossby, 1999; Woodworth et al., 2019), and wind
blowing parallel to the coast results in Ekman transport either towards or away from the coast, increasing or decreasing the sea-
level. These conditions are usually associated with storms, which is why the associated sea-level variations are called "storm
surges". For instance, in Brest (France), the amplitude of tidal variations is close to 4m, and storm surges can amount to as
much as 1.5m.

Tide gauges are numerous, forming a dense global network in recent years, and a sparser one in the last centuries. As an
example and from the GESLA-3 sea-level database (Haigh et al., 2023), 10 coastal tide gauge records start before 1907 in the
North-American east coast, and 20 start before 1900 in Europe. Old tide gauges have varying observation frequencies, from
hourly (Wöppelmann et al., 2006) to daily averages (Marcos et al., 2021). Although the sea-level measured by tide gauges is
only an indirect tracer of atmospheric pressure variability, the scarcity of direct sea-level pressure measurements motivates the
use of tide gauges to study past atmospheric fluctuations. Indeed, even when pressure measurements exist, they are often not
yet digitized and even less available in global repositories (Brönnimann et al., 2019).

It is possible to link sea-level variations with atmospheric phenomena using physical laws and models (Lazure and Dumas,
2008), or using statistical tools (Quintana et al., 2021). This work adopts the second approach, but the underlying physical
phenomena will often be used to motivate and interpret the statistical models. Multivariate linear regression (LR) will be used
to relate the storm surge to local mean-sea-level pressure anomaly and pressure gradients. Hidden Markov Models (HMM)
will allow to perform time-smoothing of probabilities given to members of 20CRv3, taking advantage of the time-continuity
of each member.

We stress that the reader should be more concerned with the underlying ideas of this work than with the particular methodol-
ogy and datasets used. Indeed, most of what is exposed here can be reiterated with other old tide gauge records, in particular in
Europe and North-America, and using other statistical tools or physical models. The Brest and Saint-Nazaire tide gauges have
the advantage of both being hourly sampled even in the 19th century, and of being close to each other. The statistical methods
were chosen for their simplicity of use and interpretation.

The data and preprocessing are detailed in section 2. Section 3 studies changes in linear relationship between storm surges
and 20CRv3 members SLP in the 19th century. Section 4 explores corrections in sea-level pressure induced by the conditional





weighing of 20CRv3 members based on surge observations. Conclusions on the proposed methodology and experiments are drawn in section 5, along with potential applications of this work.

## 2   Data

### 2.1   The Twentieth Century Reanalysis version 3 (20CRv3)

The Twentieth Century Reanalysis Project (Compo et al., 2011) aims at producing a global atmospheric reanalysis ending in 2015 and extending back to the 19th century. The present paper uses the latest version, 20CRv3 Slivinski et al. (2019), which extends up to 1806. It is an atmospheric reanalysis with 80 members, using an Ensemble Kalman Filter data assimilation scheme Evensen (2003). It assimilates only surface pressure observation, from ships and fixed stations, as well as analysed cyclone-related IBTrACs data. These surface pressure observations are taken from the International Surface Pressure Databank (ISPD) which was created for the 20CR project but also exists as an independent product Compo et al. (2019). In 20CR, the sea-surface temperature and sea-ice cover are prescribed as boundary conditions. Sea-surface temperature and sea-ice cover both benefit from satellite observations from 1981 to 2015 (the end of the reanalysis), allowing more precise boundary conditions.

The surface pressure observation density is considerably lower in the 19th century then in the late 20th century. An online platform (https://psl.noaa.gov/data/20CRv3_ISPD_obscounts_bymonth) allows to consult the monthly observation count per $2° \times 2°$ box. Fig. 1 shows yearly averages of the number of surface pressure observations per day, comparing years 1870 and 2000. The maximum value was set to 24 observations per day although in 2000 this value is mostly exceeded. In year 1870, approximately half of Europe's land surface has no observation at all, and only less than 10 points have more than 10 observations per day. Observations coming from ships allow to raise the number of observations to approximately one per day on dense traffic areas. Conversely, in year 2000, virtually all of western Europe's land has more than 24 observations per day. Taking a spatial average over the whole map from Fig. 1 gives approximately one observation every three days in 1870, versus 44 observations per day in 2000. The number of available observations is also highly variable through time, especially in the 19th century. For instance, in the $2° \times 2°$ box centered on 49°-latitude, -5°-longitude, the number of monthly observations in 1870 ranges from 2 (January, 1870) to 85 (May, 1870), while in 2000 it ranges from 2152 (June, 2000) to 3242 (May, 2000). This double constraint of data scarcity and high variations in sampling frequency in the 19th century legitimizes the search for other sources of observations to study centuries-old atmospheric variability.

Anomalies are considered with a reference climatology computed as an average over all members and over $\pm 30$ calendar days, $\pm$ 3 day hours (to include diurnal effects). In the following, we use two different periods, one is called hereafter "the satellite-era" (1980-2015) because in this period the reanalysis is forced by satellite-derived sea-surface temperature and sea-ice cover, the second is 1870-1896. A different climatology is computed for each of these two periods to retrieve anomalies.





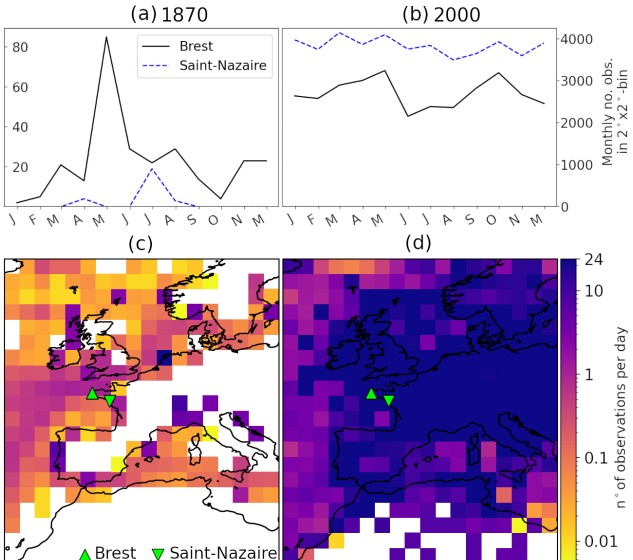

**Figure 1.** Number of surface pressure observations from the International Surface Pressure Databank (ISPD) assimilated in the Twentieth century Reanalysis version 3 (20CRv3). Top: Monthly in $2° \times 2°$ box centered on Brest and Saint-Nazaire, for years 1870 (a) and 2000 (b). Bottom: Yearly average of daily number of observation in 1870 (c) and 2000 (d).

## 2.2 Tide gauges of Brest and Saint-Nazaire (France)

In this study, the tide gauges of Brest and Saint-Nazaire are used as indirect tracers of atmospheric circulation through storm surges. The storm surges of Brest and Saint-Nazaire are highly correlated (correlation coefficient of 0.92 using the whole overlapping period). The two stations are 206km apart, which is small compared to the typical size of storms and anticyclones. The sea-levels in the two cities usually respond similarly to passing atmospheric systems. Therefore, one station can fill observational gaps of the other. However, when data from both stations is available, the distance between them is large enough to potentially get information on the direction, velocity, and size of passing systems, as the stations could provide a kind of gradient along the line between the two.

The availability of sea-level records in Brest and Saint-Nazaire in the GESLA-3 database is shown in Fig. 2. Brest's tide gauge with hourly sampling starts in 1846, while Saint-Nazaire starts in 1863. Apart from a few large gaps, both records are mostly continuous during periods 1863-1920 and 1953-present. This combination of historical and actual records is at the foundation of the methodology exposed in the next section.

## 2.3 Preprocessing of sea-level data

As mentioned earlier, the part of the sea-level which responds to atmospheric processes is the surge (also called "storm surge" or "skew surge"). To access the surge, one has to remove the tidal part of the signal, along with a possible time-linear trend in the 20th century due to sea-level rise (Cazenave and Llovel, 2010).




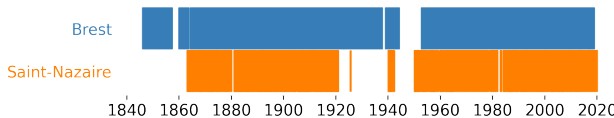

**Figure 2.** Brest and Saint-Nazaire tide gauge record availability trough time, from the GELSA-3 database (Haigh et al., 2023).

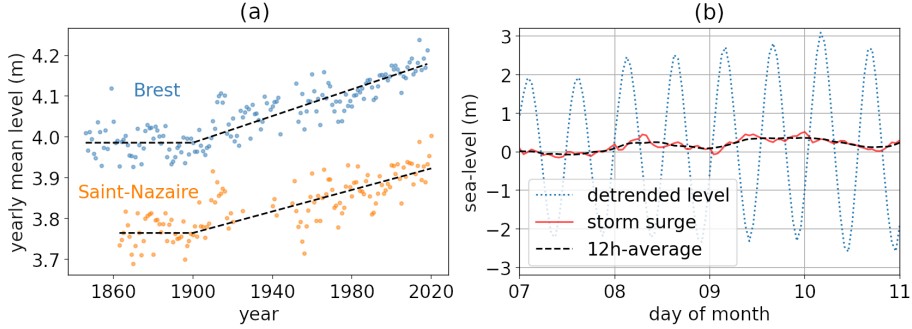

**Figure 3.** (a) Dots: Yearly averaged sea-level in Brest (blue) and Saint-Nazaire (orange). Dashed lines: constant (before 1900) and linear (after 1900) trends from least-squares fit. To improve readability, a constant +30cm was added to the Saint-Nazaire sea-level in this figure. (b) An example of detrended sea-level and associated storm surge, along with the 12-hours moving average of the surge (Brest tide gauge, March 1963).

To remove the time-linear trend in Brest's and Saint-Nazaire's signals, we first compute the yearly average of the sea-level, excluding year 1952 in Brest because it contains less than 3 month of observations which leads to an abnormally high value.

Then, we compute the average of these levels before 1900, and perform a one-parameter, time-linear, least-squares fit starting in 1900 with the value in January 1900 fixed to the 19th-century average. This gives a two-part function for each tide gauge, constant before 1900 and time-linear after 1900. Removing these functions to the signal allows to access detrended sea-levels.

Then, the tidal constituants of the detrended sea-levels are computed using U-Tide Codiga (2011), which performs harmonic (Fourier) decomposition with prescribed frequencies corresponding to planetary movements. This allows to compute and re-

move the tidal part of the sea-level, and access the surge. Then, we perform a 12h-moving average of the surge to filter out oscillations occurring at a frequency close to 12h. These oscillations are either due to tide-surge interactions (Horsburgh and Wilson, 2007) or to measurement errors in the 19th century leading to phase shifts. Furthermore, tide-surge interactions lead to stronger surges in low-tide and weaker surges in high-tide (Horsburgh and Wilson, 2007). As these phenomenons are not linked to atmospheric processes, we chose to filter them out with a simple 12h-average. This also implies that these 12-hours-averaged

surges will only respond to atmospheric events persisting for more than 12 hours.





## 3 Linear Regression (LR) between surges and sea-level pressure

### 3.1 LR in the satellite-era (1981-2015)

To estimate the statistical relationship between surges in Brest and Saint-Nazaire, and neighbouring fields of sea-level pressure, we first focus on the period 1981-2015 during which satellite data is used in 20CRv3 to constrain sea-surface temperature and
sea-ice cover, and a large number of pressure observations gives high confidence in 20CRv3 fields of pressure reduced at mean sea-level (SLP in the following).

The filtered surges described in the previous section respond to sub-seasonal variations in atmospheric pressure and winds at time scales greater than 12 hours. First, the surges in Brest and Saint-Nazaire are highly sensitive to pressure anomalies, a phenomenon called the "inverse barometer effect" (Roden and Rossby, 1999). An increase (respectively, decrease) of 1hPa
in pressure at the mean sea-level leads to a decrease (respectively, increase) in sea-level of approximately 1cm, which can be expressed as a hydrostatic equilibrium between the atmospheric and water columns.

Second, wind imposes a stress on the sea-surface, creating fluid motion through the water column in a shape known as the Ekman spiral, governed by the balance between Coriolis and turbulent drag forces. At the top of this spiral, the ocean surface current deviates at $45°$ to the right of the wind stress in the northern hemisphere. The resulting net fluid motion is called the
"Ekman transport" (Price et al., 1987), and is perpendicular to the wind stress direction, to the right in the northern hemisphere. This transport has a non-linear influences on storm surges (Bryant and Akbar, 2016; Pineau-Guillou et al., 2018).

To take advantage of these well-known physical processes, we use a simple linear regression as a statistical learning strategy, based on the 12h-averaged surges and data from 20CRv3. We take as regressors or "explanatory variables" the anomaly of pressure reduced at mean sea-level ($SLP_B$ and $SLP_{SN}$ in the following) interpolated at the coastal city of interest (Brest
or Saint-Nazaire), and the two components of a local, horizontal gradient of SLP ($\Delta_{lon}SLP_B$, $\Delta_{lon}SLP_{SN}$ and $\Delta_{lat}SLP_B$, $\Delta_{lat}SLP_{SN}$) expressed as longitudinal and latitudinal differences at $\pm 2°$ around the city of interest. In the satellite-era, we use the average SLP anomaly over all 80 members of the reanalysis, as in this period the ensemble spread in SLP between members is considerably low (less than 100Pa) due to the high observation density.

To summarize, the regressions in the satellite-era have the following form:

$$
\quad \begin{bmatrix} s_B \\ s_{SN} \end{bmatrix} \sim \mathcal{N}\left( \begin{bmatrix} \alpha_B SLP_B + \beta_B \Delta_{lat}SLP_B + \gamma_B \Delta_{lon}SLP_B \\ \alpha_{SN} SLP_{SN} + \beta_{SN} \Delta_{lat}SLP_{SN} + \gamma_{SN} \Delta_{lon}SLP_{SN} \end{bmatrix}, \begin{bmatrix} \sigma^2_{LR,B} & \text{Cov}_{B,SN} \\ \text{Cov}_{B,SN} & \sigma^2_{LR,SN} \end{bmatrix} \right) \quad (1)
$$

where $s_B$ and $s_{SN}$ are the surges in Brest and Saint-Nazaire, considered as normal (i.e. Gaussian) random variables with averages and variances given by the linear regressions estimated on the satellite-era ensemble-average SLP anomaly and respective gradients at Brest and Saint-Nazaire. It is implicit in the above formulae that $s_B$ and $s_{SN}$ are univariate variables conditioned by the values of SLP and SLP gradients. For a given map of SLP, the surges in Brest and Saint-Nazaire are assumed to be
normal random variables.



|  | $\alpha_{sat}$ | $\beta_{sat}$ | $\gamma_{sat}$ | $R^2$ |
|---|---|---|---|---|
| Brest | -1.105 | 1.202 | -0.137 | 0.834 |
| Brest 1DLR | -1.210 | 0 | 0 | 0.775 |
| Saint-Nazaire | -1.171 | 1.809 | -0.856 | 0.803 |
| St-Nazaire 1DLR | -1.466 | 0 | 0 | 0.676 |

**Table 1.** Coefficients of the linear regressions with the surges of Brest and Saint-Nazaire as explained variables and the SLP ($\alpha$) and its horizontal gradients ($\beta$ and $\gamma$) as explanatory variables. The linear regressions are computed on the satellite-era (1981-2015) ensemble-average from 20CrRv3. The results of one-dimensional linear-regressions (1DLR) are shown for comparison.

Also, we assume homoscedasticity: the variance of the surges are independent of the values of SLP, only the average depends on SLP values. These variances $\sigma^2_{sat,B}$ and $\sigma^2_{sat,SN}$ are estimated as the average of squared differences between the actual value of the surges and the SLP-based average with the LR coefficients:

$$\sigma^2_{LR} = \left\langle \left\{ s(t) - \left( \alpha \text{SLP}(t) + \beta \Delta_{lat} \text{SLP}(t) + \gamma \Delta_{lon} \text{SLP}(t) \right) \right\}^2 \right\rangle_{t \in [1981-2015]}, \tag{2}$$

where angle brackets mean average, and we have removed the indices $B$ or $SN$ to improve readability of the formula. This allows to define the coefficient of determination $R^2$ as follows:

$$R^2 = 1 - \frac{\sigma^2_{LR}}{\left\langle \left( s - \langle s \rangle \right)^2 \right\rangle}, \tag{3}$$

which can be defined for any city (Brest or Saint-Nazaire) and any period (here the averages are implicitly taken in the satellite-era). A value of 1 indicates perfect prediction, while a value of zero is obtained for a constant, average prediction.

The coefficient of determination $R^2$ can be understood as a percentage of explained variance.

The values of the LR coefficients and coefficients of determination are shown in Table 1. For comparison, a one-dimensional LR is performed with only $\alpha$ and imposing $\beta = \gamma = 0$. Fig 4(a,b) shows scatter-plots of the data used to estimate the LR around the LR mean values.

All three coefficients of these linear regressions can be understood from a physical point of view. The first coefficient, in 160 front of SLP, is the "inverse barometer". A theoretical hydrostatic equilibrium between the atmosphere and ocean would give a coefficient of -1.02, which is very close to the two coefficients found in these regressions (see Table 1). The coefficients $\beta$ and $\gamma$ of the linear regression indicate the most favorable geostrophic wind directions (Obukhov, 1962), as well as the intensity of the interaction between geostrophic wind stress and surge. This allows to estimate the most surge-favorable wind directions according to the linear regressions, as shown in Fig. 4(c). Larger coefficients in Saint-Nazaire can be interpreted as a 165 consequence of a shallower sea near Saint-Nazaire, leading to a stronger influence of winds, as can also be witnessed in other regions such as the North-Sea (Pineau-Guillou et al., 2020).





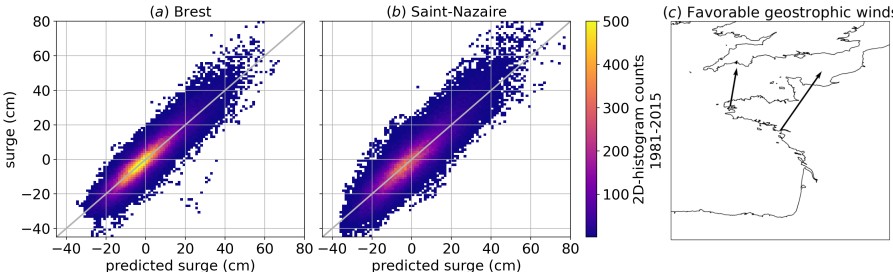

**Figure 4.** Two-dimensional histograms of surges in (a) Brest and (b) Saint-Nazaire, versus linear fit using ensemble-averaged sea-level pressure anomaly SLP, longitudinal and latitudinal gradients (at $\pm 2°$) of SLP at the city of interest. The predicted surge is $\alpha \mathrm{SLP}(t) + \beta \Delta_{lat}\mathrm{SLP}(t) + \gamma \Delta_{lon}\mathrm{SLP}(t)$ with coefficients differing for Brest and Saint-Nazaire and given in Table 1. (c) Black arrows: direction of geostrophic wind projection associated with the coefficients facing $\Delta_{lon}\mathrm{SLP}$ and $\Delta_{lat}\mathrm{SLP}$, leading to an increase in sea-level (surge). Arrows' lengths are proportional to the LR coefficients, thus large arrows indicate a stronger sensitivity of the sea-level to the winds.

For Brest, accounting for geostrophic wind only allows to explain an additional 6% of the surge variance (comparison of $R^2$ with one-dimensional linear regression, Table 1). For Saint-Nazaire, the geostrophic winds allow to explain an additional +12% of surge variance, which confirms the stronger influence of winds in Saint-Nazaire. Furthermore, although the linear

coefficient related to the inverse barometer should approach -1.02 on physical grounds, it departs from this value when using the one-dimensional LR and neglecting the influence of wind for both portal cities (Table 1). This justifies the use of a three-dimensional LR although most of the variability is explained by the first coefficient $\alpha$.

A larger spread around the linear fit is observed in Saint-Nazaire ($\sigma_{SN} = 6.8$cm) than in Brest ($\sigma_B = 5.4$cm). This difference could reflect the difference in contributions of physical phenomenon such as surface currents and wave setup, which are not

captured by the LR (by construction).

## 3.2 LR in the 19th century (1870-1896)

Is the statistical relationship between the surges in Brest and Saint-Nazaire and the SLP from 20CRv3 consistent through time ? The answer to this question is delicate, but could provide both an independent means of validation for 20CRv3 and a justification for the use of old tide gauges to probe 19th-century atmospheric variability. One way to try and answer this

question is to compare the linear relationships established in section 3.1 using satellite-era 20CRv3 member's average with the linear relationships using 19th-century 20CRv3 members.

First, one must note that the spread in SLP between 20CRv3 members is a decreasing function of time, as it is anti-correlated with the observation density from the ISPD. The statistical relationship between surges and members-averaged SLP is therefore bound to evolve through time. More precisely, the 19th-century members-average is expected to be smoother, with weaker

extreme values.

Fig. 5 provides a visual inspection as to whether the LR established in Fig. 4(a) between Brest surge and SLP fields holds in the 19th century, using either the members average (Fig. 5b) or individual members (Fig. 5c,d). In Fig. 5b, a small systematic





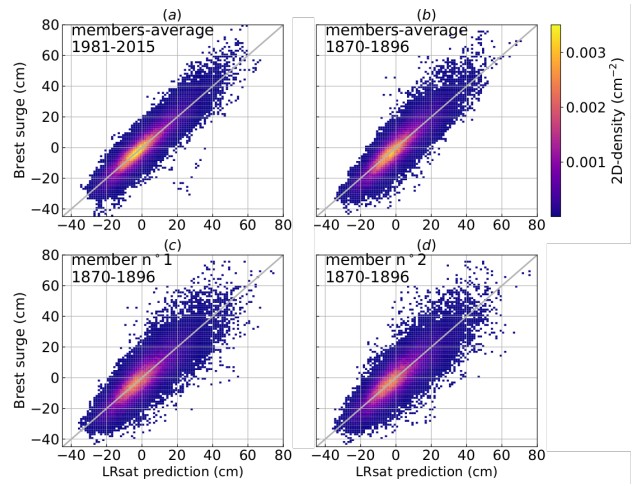

**Figure 5.** Histograms of Brest surge vs. linear combination of SLP and horizontal gradients, with the same coefficients for all sub-figures $(a - d)$. These coefficients are taken from the satellite-era LR (Fig. 4a and Table 1). (a) SLP is taken from the 20CRv3-members-average in the satellite-era as in Fig. 4a. (b) SLP is taken from the 20CRv3-members-average from years 1864 to 1896. (c) SLP is taken from the first member of 20CRv3 from years 1870 to 1896. (d) SLP is taken from the second member of 20CRv3 from years 1870 to 1896.

|  | std($\mathrm{SLP_{Brest}}$) | std($\Delta_{\mathrm{lon}}\mathrm{SLP_{Brest}}$) | std($\Delta_{\mathrm{lat}}\mathrm{SLP_{Brest}}$) |
|---|---|---|---|
| 1981-2015 (ensemble average) | 949.1 | 259.9 | 486.1 |
| 1864-1896 (ensemble average) | 911.4 (-4.0%) | 242.3 (-6.8%) | 466.7 (-4.0%) |
| 1864-1896 (individual members) | 948.7 (-0.04%) | 292.1 (+12.4%) | 517.2 (+6.4%) |

**Table 2.** Standard deviations of SLP and horizontal gradients at the city of Brest, during the two different time periods used in Fig 5, expressed in Pascals. For the period 1981-2015, ensemble average and individual members show no significant difference. Relative differences with respect to 1981-2015 are expressed in percentages.

bias is visible, the satellite-era LR consistently underestimating the amplitude moderately high and low values of surge. Indeed, for predictions around +20cm (respectively -20cm) there are more values above than below (respectively below than above) the $x = y$ diagonal line. This is consistent with the expectation that the ensemble average underestimates SLP variability due to smoothing in the 19th century. However, the LR seems to hold for low values. A negative bias is observed towards positive extremes (i.e., the highest surges are underestimated) but this phenomenon also holds with the satellite-era data. This interpretation is further confirmed by comparing standard deviations in SLP, $\Delta_{\mathrm{lon}}$SLP and $\Delta_{\mathrm{lat}}$SLP, in the period 1864-1896 and in the period 1981-2015. Table 2 shows that, when taking the 20CRv3 ensemble average in Brest, the SLP and its gradients have a lower standard-deviation in 1864-1896 compared to 1981-2015, by more than 4%.

The histograms of Fig 5(c,d) show a larger spread around the diagonal, and no clear sign of bias. This can be interpreted as the result of larger uncertainties in 1864-1896 regarding the value of the SLP in Brest due to a lower observation density. This added uncertainty leads to a higher spread in the linear relationship between SLP and surge. Table 2 indicates that the standard





deviation in time of individual members' SLP is consistent between 1864-1896 and 1981-2015 (less than 0.1% difference),
and since this first term in the LR is the main driver of the response in surge to the atmosphere one finds a similar amplitude
of linear surge-prediction based on individual members' SLP. Table 2 also indicates larger standard deviations in horizontal
gradients of SLP in the 19th century, which could be interpreted as a tendency of the model used in 20CRv3 to create stronger
gradients of SLP when not constrained by observations. In Brest, this has a minor impact on the LR between surges and SLP,
as winds only mildly contribute to the surges. However, the situation might be different when using tide gauges of shallower
seas which are more sensitive to wind conditions.

These visual inspections can be refined by computing the LR as in section 3.1 but using the data from 1864-1896 (either
ensemble average or individual members) rather than 1981-2015. The coefficients $\alpha$, $\beta$, $\gamma$ are reported in Table 3 for vari-
ous choices of SLP in years 1864-1896, compared to the coefficients from the period 1981-2015. The first coefficient, $\alpha$, is
slightly larger with the 1864-1896 ensemble-average, and slightly smaller with the 1864-1896 ensemble members, which can
be interpreted as a consequence of a smoothing in the average and an added variability in the individual members. The other
coefficients $\beta$ and $\gamma$ are harder to interpret, because they contribute less to the explained variability of the surge ($\alpha_{\text{sat}}$ alone
explains 79% of the variability in 1981-2015, thus $\beta_{\text{sat}}$ and $\gamma_{\text{sat}}$ only explain the remaining 4%), and because of the differences
in behaviour of the SLP gradients in 1864-1896 when considering ensemble averages or individual members as depicted in
table 2. However, the angles of the corresponding geostrophic wind directions with respect to a purely westerly wind are very
similar, respectively 80.3° (1981-2015), 77.5° (1864-1896, ensemble average), and 69.2° (1864-1896, individual members).
Therefore, in any situation winds blowing from the south are associated with larger surges. The amplitude of the coefficient
is similar between 1981-2015 and 1864-1896 ensemble average, while it is smaller for 1864-1896 individual members, which
can be interpreted again as a consequence of stronger longitudinal gradients of SLP for less-constrained individual members
of 20CRv3 (table 2).

Finally, coefficients of determination $R^2$ are given in table 3, using the same formula as in Eq. (3), but computin $\sigma^2_{LR}$ either
with "1864-1896" coefficients (i.e., the coefficients indicated in the second and third lines of the table) or with the satellite-era
coefficients (i.e., coefficients in the first line of the table). One can see that the LR explains 83% of the surge variability in
1981-2015, which is a quantitative view of the high correlation observed in Fig. 4(a). For the other time period 1864-1896 and
taking the ensemble average, the percentage of explained variability with the optimal choice of coefficients is 79%, which is
very close to the explained variability when using the coefficients computed in the satellite-era, indicating a good persistence in
time of the statistical LR between Brest's surge and 20CRv3's ensemble average SLP. The slight decrease in explained variance
with respect to 1981-2015 can be explained by the fact that the ensemble average SLP in 1870-1896 is less constrained by
observatons, and therefore more likely to deviate from the truth. This added variability in SLP blurs the statistical relationship
between SLP and surges. Finally, for individual members, another layer of variability is added, corresponding to the ensemble
spread, so that individual members are even further from the true 19th-century atmospheric state, resulting in an explained
variance of 69.9%. Again, however, a very similar score is explained when using the coefficients derived in the satellite-era
LR, confirming the persistence through time of the statistical LR between the Brest surge and the local SLP and its gradients.





| | $\alpha$ | $\beta$ | $\gamma$ | $R^2$ with "1864-96" coefs. | $R^2$ with "sat.-era" coefs. |
|---|---|---|---|---|---|
| 1981-2015 (ensemble average) | -1.105 | 1.202 | -0.137 | - | 0.834 |
| 1864-1896 (ensemble average) | -1.156 | 1.269 | -0.186 | -0.186 | 0.791 |
| 1864-1896 (individual members) | -1.082 ± 0.005 | 0.773 ± 0.017 | -0.194 ± 0.010 | 0.713 ± 0.003 | 0.699 ± 0.003 |

**Table 3.** Coefficients of the linear regression between, on the one hand, the surge in Brest, and on the other hand, the SLP (coefficient $\alpha$) and its horizontal gradients (coefficients $\beta$ and $\gamma$). The satellite-era coefficient are compared with coefficients in 1870-1896, with values for the SLP taken either as the ensemble average of 20CRv3 members, or individual members. For the individual members, values given are in the format : average $\pm$ standard-deviation over the 80 members. Coefficients of determination $R^2$ are given in two case scenarios: the true surge value is compared to its predicted value either according to the LR with actual coefficients $\alpha$, $\beta$, $\gamma$ on the corresponding line, or according to the LR with satellite-era coefficients $\alpha_{\text{sat}}$, $\beta_{\text{sat}}$, $\gamma_{\text{sat}}$ (from the first line 1981-2015).

The same behaviour can be witnessed for Saint-Nazaire (not shown for conciseness). This investigation of the LR between observed surges and 20CRv3-SLP first shows that the statistical relationship between these two quantities appears to remain valid in the 19th century, which is by itself a validation result for 20CRv3 using an independent, indirect source of observation of atmospheric variability. Second, it appears that the added variability observed in the end of the 19th century is not due to a modified linear relationship, but rather to an increase of variability in 20CRv3. The fact that this increase is visible in the linear projection on the surges of Brest and Saint-Nazaire indicates that the surges of Brest and Saint-Nazaire contain valuable information on the atmospheric variability that could be used to reduce the uncertainties in 20CRv3. To reformulate, this analysis suggests that the surges contain information on the 19th-century atmospheric variability that is complementary with the observational information gather in 20CRv3 through surface pressure observations from the ISPD.

The next section explores the potential to use Brest and Saint-Nazaire surge observations to constrain the 20CRv3 through weighing of its members.

## 4 Probability of 20CR members based on surge observations

### 4.1 Hidden Markov Model (HMM)

In the 19th century, the spread between 20CRv3 members is much larger than in the period 1981-2015, and one of the aims of this work is to estimate conditional probabilities of each member of the reanalysis, based on surge values in Brest and Saint-Nazaire. Note that in the reanalysis, the members are assumed to have uniform probabilities, that is a probability of 1/80, since we have 80 members.



Therefore, one aim of this work is to evaluate the change in probability distribution of the members when adding the information of the surges in Brest and/or Saint-Nazaire. One can estimate conditional probabilities of each member at time $t$ based on the values of the surges at time $t$. To do that, we use the satellite-era-derived linear regression expressed in section 3.1. Using Baye's theorem, we know that the probability to observe a SLP map conditionally on observing a surge value is proportional to the probability of observing a surge given the SLP map, which we assume can be expressed simply by the

satellite-era LR.

To differentiate these member probabilities from the ones we will derive later on using a hidden Markov model, we use the notation $p_{\text{HMM}}(i,t)$ for the probability of member $i$ at time $t$.

$$\begin{cases} p_{\text{HMM}}(i,t) \propto p_{\text{sat}}\left(s - [\alpha_{\text{sat}}\text{SLP}_i(t) + \beta_{\text{sat}}\Delta_{\text{lon}}\text{SLP}_i(t) + \gamma_{\text{sat}}\Delta_{\text{lat}}\text{SLP}_i(t)]\right), \\ \sum_{i=1}^{80} p_{\text{HMM}}(i,t) = 1. \end{cases} \quad (4)$$

Although these probabilities already bear significant information, they have the undesirable property to be time-discontinuous.

This is not coherent with the fact that the members of 20CRv3 are time-continuous: they are propagated in time using a NWP model. To remedy this issue, we compute smoothed (or reanalyzed) probabilities using a hidden Markov model (HMM) detailed below, which we write $p_{\text{HMM}}(i,t)$:

$$p_{\text{HMM}}(i,t) := P\left(\text{member}(t) = i \;\middle|\; \begin{bmatrix} \text{surge}(t=1) \\ \vdots \\ \text{surge}(t=T) \end{bmatrix} = \begin{bmatrix} s_1 \\ \vdots \\ s_T \end{bmatrix}\right), \quad (5)$$

where one uses an observational record of surges from time-index $1$ to $T$. This quantity is a time-smoothed version of $p_{\text{HMM}}(i,t)$

which takes into account past and future values of the surge. For this purpose, a simple Hidden Markov Model (HMM) is used. The first ingredient of the HMM is the transition matrix $\mathcal{T}_{ij}(t)$ from member $i$ at time $t-1$ to member $j$ at time $t$.

$$\mathcal{T}_{ij}(t) := P\left(\text{member}(t) = j \mid \text{member}(t-1) = i\right). \quad (6)$$

To estimate the transition matrix, a strong hypothesis is made:

$$\mathcal{T}_{ij}(t) \propto K_\theta\left(SLP_{\text{map},j}(t), SLP_{\text{map},i}(t)\right), \quad (7)$$

where $SLP_{\text{map},i}(t)$ is the $i$-th member's map of mean-sea-level pressure in a squared box of $18°\text{W} \leq \text{lon} \leq 18°\text{E}$, $28°\text{N} \leq \text{lat} \leq 64°\text{N}$ at time $t$, and $K_\theta(\cdot,\cdot)$ is a positive real-valued function that measures the similarity between $SLP_{\text{map},i}(t)$ and $SLP_{\text{map},j}(t)$ and depends on parameters $\theta$. Note that in case of missing observations between $t$ and $t+q-1$, where $q \geq 1$, we assume the following form for the more general transition matrix $\mathcal{T}'$:



$$\mathcal{T}'_{ij}(t,q) := P\left(\text{member}(t+q) = j \mid \text{member}(t-1) = i\right) , \tag{8}$$

$$\propto \left\{K_\theta\left(SLP_{\text{map},j}(t+q), SLP_{\text{map},i}(t+q)\right)\right\}^{\frac{\tau}{\tau+q}} , \tag{9}$$

where $\tau > 0$ is a hyperparameter. We set $\tau$ to 16 in time-step unit, and since we use time-steps of 3 hours, our $\tau$ corresponds to a time of 2 days. When $q$ is large compared to $\tau$, the coefficients of the transition matrix become independent of $i, j$ and transitions from any state to another are equally allowed (i.e., memory is lost).

Eq. (7) states that transitions from one member to another are more likely if the associated SLP map at time $t$ are similar. This prevents abrupt transitions to dissimilar atmospheric states. The size and location of the map was chosen to cover an area inside which storms and anticyclones which affect the surges in Brest and Saint-Nazaire would lie. Ideally, $K_\theta(\cdot, \cdot)$ should be symmetric, semi-definite. Here, a simple Gaussian kernel of Euclidean distances is used, with normalization factor $\theta > 0$, so that for two fields $X$ and $Y$:

$$K_\theta\left(X, Y\right) = \exp\left\{-\sum_{n \in \text{lons}} \sum_{l \in \text{lats}} \frac{(X_{nl} - Y_{nl})^2}{\theta^2}\right\} , \tag{10}$$

where the sum over $n$ and $l$ represents a sum over longitudes and latitudes. We then define $\Theta$ through the following equation:

$$\frac{\theta}{\Theta} = \overline{\left(\frac{1}{80}\sum_{i=1}^{80}\frac{1}{T}\sum_{t=0}^{T} SLP_{\text{map},i}(t)^2\right)^{1/2}} , \tag{11}$$

where the overbar denotes spatial average. This normalization will allow to optimize $\theta$ through grid search of $\Theta$ for a maximum of likelihood of the surge observations.

One can compute $\mathcal{T}_{ij}(t)$ by setting a value of $\theta$ and using the hypothesis of Eq. (7) along with the fact that for all $i, t$, we have $\sum_j \mathcal{T}_{ij}(t) = 1$. This then allows to estimate $p_{\text{HMM}}(i, t)$ with the forward-backward algorithm (Rabiner, 1989). Let:

$$a_i(t) := P\left(\begin{bmatrix}\text{surge}(1)\\ \vdots \\ \text{surge}(t)\end{bmatrix} = \begin{bmatrix}s_1 \\ \vdots \\ s_t\end{bmatrix} , \text{m}(t) = i\right) , \tag{12}$$

$$b_i(t) := P\left(\begin{bmatrix}\text{surge}(t+1)\\ \vdots \\ \text{surge}(T)\end{bmatrix} = \begin{bmatrix}s_{t+1} \\ \vdots \\ s_T\end{bmatrix} \middle| \text{m}(t) = i\right) . \tag{13}$$

These two quantities can be computed recursively, following the forward procedure:





$$a_i(1) = p_{\text{HMM}}(i,1), \tag{14}$$

$$a_i(t+1) = p_{\text{HMM}}(i,t+1) \sum_{j=1}^{80} a_j(t) \mathcal{T}_{ji}(t), \tag{15}$$

and the backward procedure:

$$b_i(T) = 1, \tag{16}$$

$$b_i(t) = \sum_{j=1}^{80} b_j(t+1) \mathcal{T}_{ij}(t) p_{\text{HMM}}(j,t+1). \tag{17}$$

Finally, this allows to estimate $p_{\text{HMM}}(i,t)$ by noting that:

$$p_{\text{HMM}}(i,t) = \frac{P\left(\text{member}(t) = i, \begin{bmatrix} s(1) \\ \vdots \\ s(T) \end{bmatrix} = \begin{bmatrix} s_1 \\ \vdots \\ s_T \end{bmatrix}\right)}{P\left(\begin{bmatrix} s(1) \\ \vdots \\ s(T) \end{bmatrix} = \begin{bmatrix} s_1 \\ \vdots \\ s_T \end{bmatrix}\right)}, \tag{18}$$

which gives, in terms of $a_i(t)$ and $b_i(t)$:

$$p_{\text{HMM}}(i,t) = \frac{a_i(t)b_i(t)}{\sum_{j=1}^{80} a_j(t)b_j(t)} \tag{19}$$

while keeping in mind that Eq (19) implicitly relies on hypothesis (7) and a fixed form of $K_\theta$.

Comparing $p_{\text{HMM}}(i,t)$ with the uniform distribution $p(i,t) = \frac{1}{80}$ allows to see if the surge observations are coherent with the SLP fields of 20CRv3 (section 4.2) and to select the most relevant members given surge data (section 4.3).

To choose the parameter $\theta$, we performed a grid-search of its normalized form, $\Theta$, computed the log-likelihood of the surge observations as an output of the algorithm. Indeed, the log-likelihood $l_\theta(0 \ldots T)$ is expressed as follows:

$$l_\theta(0 \ldots T) = \log\left(\sum_{i=1}^{80} a_i(T)\right). \tag{20}$$

Figure 6 shows variations of this quantity with $\Theta$, for one year (1885) of surge observations in Brest (i.e. $t = 0$ is 01 January 1885 and $T$ is 01 January 1886). The curve shows a distinct maximum around $\Theta \approx 0.09$, and plateaus for higher values.



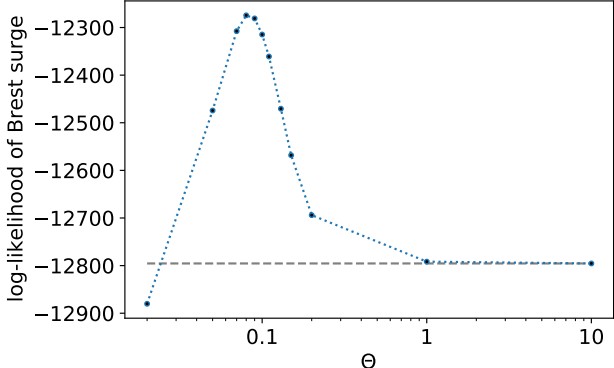

**Figure 6.** Log-likelihood of Brest surges (dotted blue line) as a function of parameter $\Theta$ defined in Eq. (11). For comparison, the log-likelihoood of the simple model without Hidden Markov Model ($\Theta \to +\infty$) is also shown (dashed grey line). The log-likelihood was estimated using data from year 1885 for a first estimation of optimal parameter $\Theta$. Values of $\Theta$ used for estimation are 0.02, 0.05, 0.07, 0.08, 0.09, 0.1, 0.11, 0.13, 0.15, 0.2, 1, and 10, with 0.09 giving the largest log-likelihood.

According to Figure 6, the difference of log-likelihood between the model without HMM ($\theta = +\infty$) and with HMM is close to 1000. The introduction of one extra parameter in the filtering model compared to the static one is thus clearly justified if the two models are compared using standard criteria usch as AIC, BIC or likelihood ratio tests Zucchini et al. (2017).

Note that in the limit $\Theta = +\infty$, we have a constant transition probability $\mathcal{T}_{ij}(t) = \frac{1}{80}$ and $p_{\text{HMM}}(i,t)$ reduces to $p_{\text{HMM}}(i,t)$.
Figure 6 thus supports the use of the HMM to estimate probabilities of SLP map conditioned by surge observations.

The choice of restricting the estimation of log-likelihood to one arbitrary year (1885) is supported by the fact that estimation of $\mathcal{T}_{ij}(t)$ is computationally expansive. We assume that the optimal value of $\theta$ generalizes well to other years. A better optimization of $\theta$ would necessitate further work that is out of the scope of this study. Setting $\Theta = 0.09$ will already enable us to find interesting features of $p_{\text{HMM}}(i,t)$.

**4.2   Modification of 20CRv3 ensemble when accounting for surges**

This section is devoted to the study of $\delta\mu_{\text{HMM}}(t)$, the difference between weighted and unweighted ensemble average, defined by:

$$\delta\mu_{\text{HMM}}(t) := \sum_{i=1}^{80} \left( p_{\text{HMM}}(i,t) - \frac{1}{80} \right) SLP_{\text{map},i}(t) \,, \tag{21}$$

where $SLP_{\text{map},i}(t)$ is a short notation for the sea-level pressure field of 20CRv3's $i$-th member. $\delta\mu_{\text{HMM}}(t)$ is defined equiva-
lently using $p_{\text{HMM}}(i,t)$. This quantity shows how strong is the average deviation when taking into account surge observations. It will also be sometimes normalized by $\sigma_{20CR}(t)$, the estimated standard deviation of the unweighted ensemble:





$$\sigma_{20\text{CR}}(t) := \left[ \frac{1}{79} \sum_{i=1}^{80} \left( SLP_{\text{map},i}(t) - \frac{1}{80} \sum_{i=1}^{80} SLP_{\text{map},i}(t) \right)^2 \right]^{1/2}. \tag{22}$$

Note that in this definition, $\sigma_{20\text{CR}}(t)$ depends on time, latitude and longitude. Therefore at each grid point and for each time step the quantity $\delta\mu_{\text{HMM}}(t)$ will be normalized by a different value, indicating the strength of the reanalysis ensemble spread at this location in time and space.

To further interpret the result of our HMM algorithm, we introduce the filtered effective ensemble size $\nu_{\text{HMM}}(t)$ (Liu, 1996):

$$\nu_{\text{HMM}}(t) := \frac{1}{\sum_{i=1}^{80} p_{\text{HMM}}(i,t)^2}, \tag{23}$$

and we define equivalently $\nu_{\text{HMM}}(t)$. These quantities are estimates of the number of ensemble members that can be retained according to surge observations, assuming one discards very unlikely members.

In Fig. 7, variables $\delta\mu_{\text{HMM}}$, $\delta\mu_{\text{HMM}}/\sigma_{20\text{CR}}$ and $\nu_{\text{HMM}}$ are shown as a function of time for the period 1846-1890. All these quantities show a strong seasonality. This is due to a much stronger SLP variability in winter, and a corresponding stronger response of the surges. The figure shows that the amount of correction $\delta\mu_{\text{HMM}}$ and the decrease in ensemble size $\nu_{\text{HMM}}$ are much stronger using smoothed probabilities with HMM rather than probabilities without HMM. Showing the deviation $\delta\mu_{\text{HMM}}$ in the Bay of Biscay, where the standard deviation of $\delta\mu_{\text{HMM}}/\sigma_{\text{HMM}}$ is strongest (see Fig. 8), substantial absolute values of ∼600 Pascals are obtained in early 1850s winters, even after averaging over 3 months. These large deviations correspond to more than one standard deviations of the ensemble size. Using probabilities without HMM, deviations are weaker but still non-negligible ($\sim 500 Pa$, $\sim 0.7\sigma$). The slow decrease in $\delta\mu_{\text{HMM}}$ with time is coherent with slowly increasing observations used in 20CRv3, although with substantial decadal variations. However, $\delta\mu_{\text{HMM}}/\sigma_{\text{HMM}}$ and $\delta\mu_{\text{HMM}}/\sigma_{\text{HMM}}$ do not show a clear trend, indicating a persisting gain in information from surge observations throughout the 19th century.

In terms of effective size, Fig. 7 shows that the smoothing HMM algorithm imposes a strong member selection, with mostly only 1 member retained at each time step, in winter and before 1880. Probabilities without HMM mostly retain more than half of the members, although peak low values of $\nu_{\text{HMM}}(t)$ show that even without the HMM sometimes more than half of the ensemble members are highly unlikely. Filtered effective ensemble size reaches very low yearly and seasonal average values, indicating that many 20CR members are highly unlikely with respect to surge estimates from tide gauge observations. A strong increase in $\nu_{\text{HMM}}(t)$ is witnessed around year 1880. This can be explained by the availability of a large number of weather station data in Eastern Europe and Russia from 1880-on, and by an intensification of maritime traffic around 1880.

The spatial structure of $\delta\mu$ is examined in Fig. 8. The analysis of time-standard-deviation of $\delta\mu_{\text{HMM}}$ and $\delta\mu_{\text{HMM}}/\sigma_{20\text{CR}}$ shows that the area of greatest influence of the corrections from surge-smoothing from Brest and Saint-Nazaire tide gauges is in the Bay of Biscay. This can be explained by the passage of strong storms in the Bay of Biscay, which can cause high surges in Brest and Saint-Nazaire, and by the sparsity of direct pressure measurements (ship logs) in this area in the 19th century. Standard deviation of $\delta\mu_{\text{HMM}}$ shows largest values to the north-west of the map, which is where strong storms travel. Indeed,



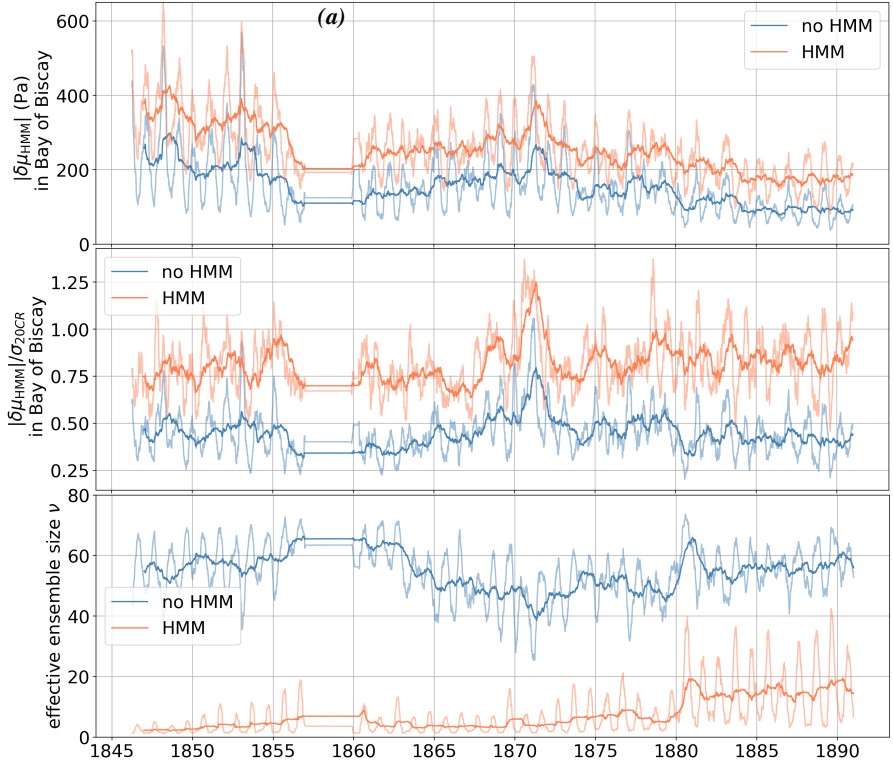

**Figure 7.** (a) Average SLP deviation $\delta\mu_{\text{HMM}}$ in Pascals, in the center of the Bay of Biscay. (b) Same as (a) normalized by reanalysis ensemble standard deviation $\delta\mu_{\text{HMM}}/\sigma_{\text{20CR}}$. (c) Effective ensemble size $\nu_{\text{HMM}}$. Orange: using smoothed probabilities with HMM $p_{\text{HMM}}(i,t)$. Blue: using probabilities without HMM $p_{\textbf{HMM}}(i,t)$. Bold: yearly average. Thin: 3-month average. All plots make simultaneous use of data from Brest and from Saint-Nazaire tide gauges.

the variability of SLP shows a great north-west gradient, as can be seen from maps of time-standard-deviation of 20CRv3 mean SLP (not shown). Noticeably, the size of the area of influence of $\delta\mu_{\text{HMM}}$ is smaller in 1880-1890, which can be explained by a greater conditioning of 20CRv3 members by observations, both offshore and in-land. In case of very sparse observations used in 20CRv3, the area of influence of these corrections widens due to continuity of SLP fields. Note, as well, that the area of influence is greater for $\delta\mu_{\text{HMM}}$ then for $\delta\mu_{\textbf{HMM}}$, because of the time-propagation of corrections thanks to the smoothing HMM algorithm. Finally, this figure confirms the great difference in amplitude of deviations between pre-1880 and post-1880 corrections, already witnessed in Fig. 7. Similar spatial footprints can be witnessed from maps of high and low quantiles of $\delta\mu$, only with different values (not shown). Similarly, computing the time-standard deviations as in Fig. 8 but restricting the times used for computation to April-September rather than October-March shows the same spatial pattern but with much lower values (not shown).

These corrections also have a strong decadal variation, with non-trivial yearly averages persisting for several years, as shown in Fig. 9. The same behaviour can be witnessed for the surge, which is strongly anti-correlated to these deviations (Fig. 9).



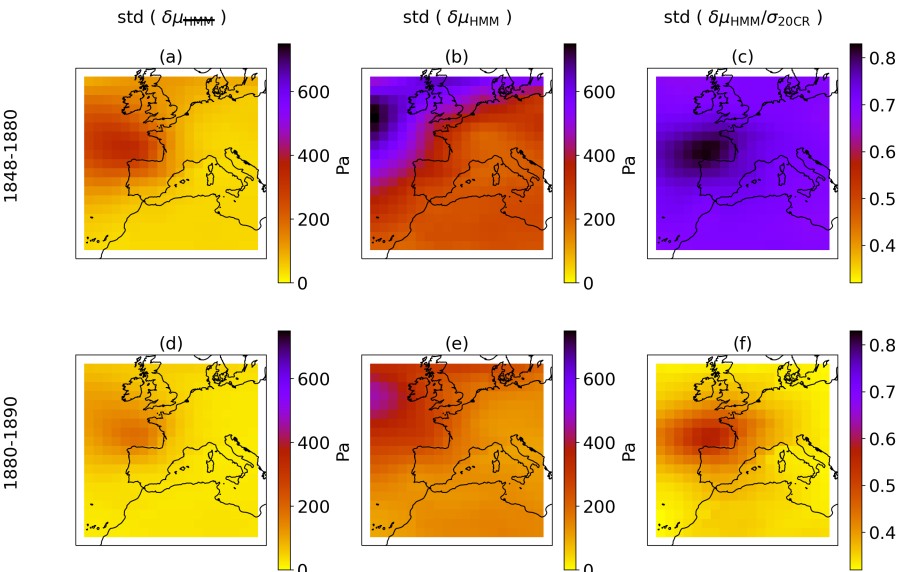

**Figure 8.** Time-standard deviation of $\delta\mu_{\text{HMM}}$ (a,d), $\delta\mu_{\text{HMM}}$ (b,e) and $\delta\mu_{\text{HMM}}/\sigma_{20\text{CR}}$ (c,f) computed from October to March, for years 1848-1880 (a,b,c) and 1880-1890 (d,e,f). Probabilities $p_{\text{HMM}}$ and $p_{\text{HMM}}$ make use of data from both Brest and Saint-Nazaire tide gauges.

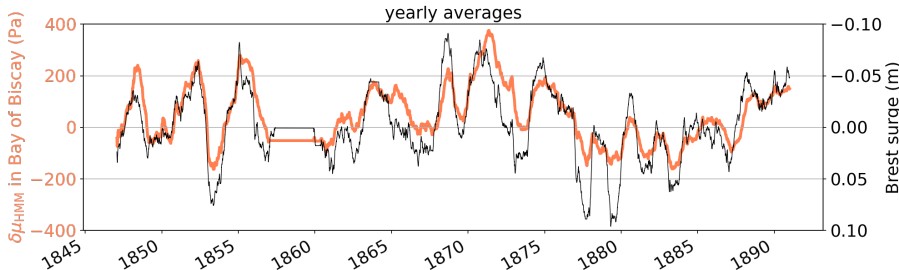

**Figure 9.** Orange, bold: yearly-average of $\delta\mu_{\text{HMM}}$ at position 47.83°lat, -7.57°lon, in the Bay of Biscay, using data from both Brest and Saint-Nazaire tide gauges. Black, thin: Brest surge (inverted sign).

This can be explained by the fact that 20CRv3 smooths SLP values in areas of sparse measurements, and that surge-filtering

corrections allow to retrieve more realistic intense values (either positive or negative). This interannual variability is related to the variability in storminess (Bärring and Fortuniak, 2009).

## 4.3 Focus on two 19th-century storms

One of the aim of this study is to show that old tide gauge data can be used to better understand past severe storms. In this section, two known storms are studied for illustrative purposes.



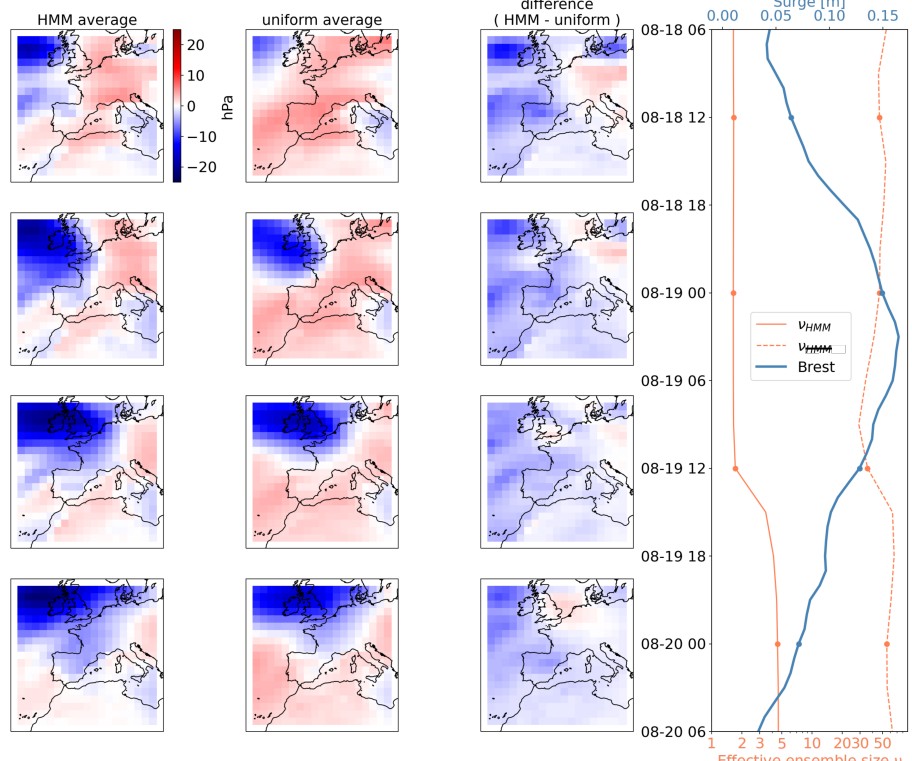

**Figure 10.** Mean result of the smoothing procedure during the Moray Firth Fishing disaster that happened during the night between august 18th and 19th, 1848. Left column: average according to surge observations and HMM smoothing algorithm (probabilites $p_{\mathrm{HMM}}(i,t)$). Middle column: average using constant uniform weights on 20CRv3 members. Right column: difference HMM - uniform. Right panel: surge observations and effective ensemble sizes. The dates of the left-hand-side plots are indicated with dots on the right-hand-side figure.

The "Moray Firth fishing disaster" of august 1848 was the result of a storm hitting the Northeastern coast of Scotland (Coull, 1995). The day before the storm, conditions seemed favorable for fishing, therefore the unexpected storm caused more than 100 casualties among Scottish fishermen, which later led to improvements in harbours and vessels in Scotland. This severe storm, although located hundreds of kilometers away from Brest, left a trace on the sea-level at the Brest tide gauge. Our estimation of surge shows a 12h-averaged positive surge of 15cm on the night between the 18th and 19th of august (Fig. 10). Probabilities without HMM give an effective ensemble size down to $\nu_{\mathrm{HMM}} \approx 30$ at 9am on august 19th, and smoothed probabilities give $\nu_{\mathrm{HMM}} \lesssim 1.7$ until 12am on that same day, indicating a strong selection among candidate SLP patterns given by members of the 20CRv3 reanalysis. The retained members give an average storm of greater intensity and spatial extent over the British Isles than using the uniform ensemble average from 20CRv3.

Although these results do not allow to assert firm statements about this storm, they show that even tide gauges located hundreds of kilometers away from the core of a pressure system can bring interesting pieces of information.



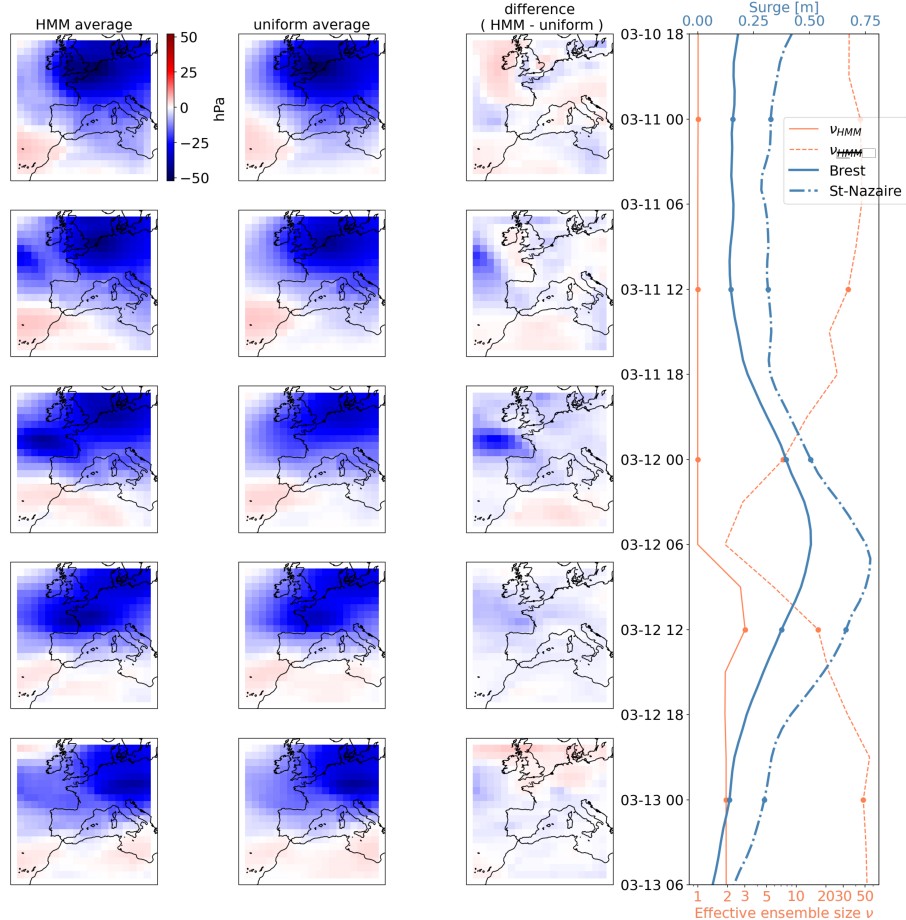

**Figure 11.** Same as Fig. 10 a storm that hit western Europe around the 12th of March 1876, later called Lothar's big brother. In this figure tide gauge observations from Saint-Nazaire are also available.

The second storm shown in Fig. 11 from the middle of march 1876 was later called Lothar's big brother in reference to the storm of Lothar, 1999. This storm induced maximum wind speeds of over 150km/h across Northern France, Belgium and Germany (Zimmerli and Renggli, 2015). As shown in Fig. 11, this storm was more intense than the one from Fig. 10, and hit directly the Northern coast of France, leaving a strong footprint on Brest and Saint-Nazaire tide gauges. Peak positive 12h-
averaged surges were of more than 50cm for Brest and close to 80cm in Saint-Nazaire. At this moment even $\nu_{\text{HMM}}$ shrinks down to $\approx 1.9$, while $\nu_{\text{HMM}} \approx 1$ more than two days before the extreme surge event. Therefore, the filtered averages on the left correspond to one single member from 20CRv3. Examining the difference $\mu_{\text{HMM}}$ on the 11th (12am) and the following night (12th, 00am), it seems that the spatial extent of the storm off the Western coast of France was underestimated in most 20CRv3 members, according to surge observations in Brest and Saint-Nazaire. It shows again that tide gauges could be used to better
estimate past atmospheric states, and in particular severe storms.



# 5   Conclusion and perspectives

This study is a proof of concept for the use of century-old tide gauge data as a means of understanding past atmospheric subseasonal variability. Storm surges of Brest (starting in 1848) and Saint-Nazaire (starting in 1863) allow to assess part of the atmospheric variability that was uncaught in global 20CR reanalyses based on pressure observations only. Weighing 20CR

members according to surge observations reduces the effective ensemble size, and implies significant deviations in members-averaged sea-level pressure in the Bay of Biscay. Through the second-half of the 19th century, these deviations diminish and the effective ensemble size rises, however they remain non-negligible.

This work has several potential applications. First, replicating this work with other tide gauges could help to validate reanalyses like 20CRv3 against independent data, and to potentially identify anomalous trends or wrong estimation of specific

events. Second, tide gauge data could be directly used to constrain atmospheric reanalyses, for instance as "inverse barometers", only with higher uncertainties than direct pressure sensors. Tide gauges which are more sensitive to wind than pressure, such as the ones in the North Sea, would require a more complex treatment, but could still be used to constrain atmospheric reanalyses. Third, tide gauges could be used to constrain regional scale atmospheric simulations in order to better estimate the magnitude and spatial extent of known past severe storms. Fourth, tide gauge records could be combined with direct ob-

servations of atmospheric pressure to give statistical estimates of atmospheric fluctuations in the 19th century without the use of a NWP model, such as the optimal interpolation of Ansell et al. (2006) based on direct pressure observations only, or the analogue upscaling of Yiou et al. (2014) for the short period 1781-1785 of dense observations in western Europe. Finally, this work could be replicated in a more general context, using other types of variables and observations, learning the relationship between observations and large-scale features using recent observations and precise reanalyses, and applying these statistical

relationship in the past to uncover past large-scale events.

*Author contributions.*   Paul Platzer wrote and run the codes, created the figures, and wrote the original manuscript. Pierre Tandeo proposed major modifications to the manuscript. Pierre Ailliot helped design the Hidden Markov Model algorithm. All authors helped conceptualizing the work, reviewed the manuscript and approved its final form.

*Competing interests.*   The authors declare that they have no competing interests.

*Acknowledgements.*   This work was supported by European Research Council (ERC) Synergy grant STUOD – DLV-856408. Support for the Twentieth Century Reanalysis Project version 3 dataset is provided by the U.S. Department of Energy, Office of Science Biological and Environmental Research (BER), by the National Oceanic and Atmospheric Administration Climate Program Office, and by the NOAA Physical Sciences Laboratory.



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
