# Peer review of "Could old tide gauges help estimate past atmospheric variability?"

_EGUsphere, 2023_

## Referee Comment (RC2)

This work uses historical tide gauge records in the French coast as proxy data to improve the atmospheric 20[th] century reanalysis. Specifically, the sea level observations are used to constrain the number of ensemble members in the reanalysis, selecting and weighting them according to their correspondence to observed storm surges. This is an interesting idea. The approach discards model realisations that, although physically consistent with the scarce available mean sea level pressure observations, are inconsistent with observations of storminess from coastal tide gauges. I am not a statistician and therefore I am not qualified to comment on the details in section 4. My comments are focused on the results and, mostly, on sections 2 and 3 on data description and processing. I have found major issues in these sections, including a basic conceptual misunderstanding of the generation of storm surges. Also, there are missing details in data processing and I have concerns on the simple linear regression model that has been used to relate storm surges and sea level pressure anomalies.

Another major concern is that the authors state in the conclusions that they use the tide gauge time series as a barometer record not assimilated in the reanalysis. However, they model the storm surges from sea level accounting for both pressure and geostrophic winds. I do not understand the interest of including the winds here (which by the way are not modelled correctly, in my opinion). If the purpose is to incorporate only pressure-like measurements, then pressure gradients should not be included at all. Or at least, the pressure-like record should be reconstructed without accounting for wind effects.

All these issues are detailed in the following. My recommendation is that the manuscript should undergo major revisions before being considered for publication.

Section 2: This section needs further details in the description of the data set. Preprocessing of data needs improvement.

- L.62: define the spatial and temporal resolution of the reanalysis.
- L. 70: I assume the variable used is mean sea level pressure (MSLP), correct?
- L. 81-82: please, remove.
- L. 83: anomalies of what? I assume this means that mean SLP anomalies are computed with respect to climatologies for 2 periods. Please, specify.
- Description of tide gauges (section 2.2): please, define the temporal sampling of tide gauges, and their overlapping periods. The statement that one time series can be used to fill in gaps of the another is incorrect. They are highly correlated at low frequencies but not necessarily at high frequencies (i.e. storms). Actually, they are not even in the same location, as stated later in the text, so differences in storm surges may be expected.
- L. 97: actual-> modern, current, present-day?
- The removal of MSL effects in the tide gauge processing is incorrect. The impact of MSL should not be removed in this way, because there are other MSL variations at interannual and decadal time scales that would remain. A better approach is to remove mean (or median) yearly averages. It also avoids the problem of selecting an arbitrary change point to calculate different linear trends.
- L. 108-109 on detiding: which is the period used for detiding?
- L. 113: phenomenons-> phenomena
- L. 110: A 12-h filtering removes part of the storm surge variability, especially that related to storms.

- L. 114-115: smaller scale events are unlikely to be recorded in 20cR at 2deg resolution.

Section 3: This section contains fundamental errors in the understanding of the processes that generate storm surges. Also, the linear regression model is questionable as applied here, and should be better justified. More details:

- L. 123-124: hydrostratic equilibrium is a general law not only affecting this region. Inverted barometer has been widely applied everywhere.
- L. 127-131: this is incorrect. The main mechanism of wind as storm surge generation is not the Ekman transport, but the piling up of waters due to wind blowing perpendicular to the coast. Check your own results, where you can actually see it (e.g. line 163). Please, change this in the abstract too.
- L. 1334-134: I assume that SLP here (and used later in equation 1) is the SLP anomaly with respect to time at the given grid point. If this intend to represent the inverted barometer effect, this is incorrect. The anomaly should be computed with respect to the average mean sea level pressure over the world oceans in the reanalysis. See for example Ponte 1994 (https://doi.org/10.1029/94JC00217).
- L. 134: interpolated how? Linearly?
- Equation 1: please, define all the terms. I do not believe this is a good way to separate the forcings. I think it would be better to use perpendicular and parallel to the coast MSLP gradients (geostrophic winds). In this case, the parallel will be insignificant and the model would be simplified. This separation has a physical meaning and would also allow to interpret the results easily.
- Also, why is the term Cov introduced? How does it change the results?
- Still in the same equation, are all coefficients significant? what happens if a step-wise regression is used instead? I would expect gamma coefficients to be discarded.
- Tables 1 and 2: units are missing.
- L. 170: "1.02 on physical grounds"-> actually not, the inverted barometer effect is a simplification and does not work at the coast.
- L. 171-172: "This justifies…"-> I do not think this statement is correct. I believe a step-wise regression would tell you if the use of these parameters is justified or not. Here it has not been proved, not even with uncertainties of the parameters.
- L. 174: wave setup would be removed with the 12-h averaging, so it does not play any role.
- Section 3.2: To demonstrate that the same coefficients can be used during another period, this is not the correct approach. I think a better approach would be to use the recent period to generate 19th-century-like observations, i.e., downgrading the number and location of observations, and compute the coefficients of the LR. These downgraded, coarse-resolution coefficients would be directly comparable to the ones in the section above with the full resolution. It would be important to calculate uncertainties in all cases in order to compare the coefficients.
- Fig 5: it is unclear what is shown here. Are coefficients of the modern period used here? Please, specify. The description of the results in fig 5 is confusing and does not reflect, in my opinion, the results. For example, in l. 188 the bias referred to one of the cases is visible in all (5a and 5b); l. 193: which interpretation is referred to here?; l. 196 states that there is no clear sign of bias in individual members but I think the bias is similar to that in the ensemble average. The bias is not only due to ensemble mean but also to limitation of coarse resolution data.

- L. 206-220: this discussion on the differences in coefficients should consider uncertainties to ensure that the use of two different periods lead to different values. Values slightly larger/smaller do not provide confidence in the results. Therefore, the interpretations are not reliable (e.g. differences attributed to ensemble averaging).
- L. 225-226: I think the consistency of the coefficients and the explained variance in modern and old periods is a consequence of the dominance of the inverted barometer effect. This is probably not true for extreme values generated by strong winds associated to storms, but it holds for mean storm surge variability driven by pressure changes.
- Summarising, I believe the LR model should be modified to consider winds parallel and normal to the coast in each case, and use step-wise (or similar) approach to remove the terms that do not explain more variance but introduce noise. Also, uncertainties of the parameters should be calculated. The use of the coefficients calculated in the modern period should be tested in downgraded modern data to prove that they are usable in older periods. The entire comparison and discussion of the two periods should be modified accordingly and simplified in case this is proved.

Section 4: I am not an expert in HMM and I am not qualified to comment on the details of the method explained here. It would be good that an expert statistician reviews this part. I nevertheless would recommend defining all variables in the equations, as well as the acronyms (e.g. NWP), as it makes it very difficult to follow as it is now. Other comments follow:

- L. 270: this is a rather large area. Are the results sensitive to this choice?
- L. 336-337: does this mean that the storm surges constrain the data more in winter when they have a stronger signal?
- L. 361: then->than
- Section 4.3 is very illustrative of the potential of the approach. However, more information would be useful to understand why including a single (or two) new records essentially rules out 79 out of 80 model ensemble members. In particular, how many sea level pressure records were included in this period (or shortly before the storms) in the area? How are they distributed? It would be also useful to see a case with more sea level pressure observations (early or mid- 20th century for example).

Conclusions section:

- L. 405-408: I understood from the LR model that winds were incorporated into the model as SLP gradients. Then, why if wind is an important driver of storm surges may limit the use of the tide gauge? In fact, if wind is not accounted for and wants to be removed, then the LR model could include only the inverted barometer effect with the adjusted parameter at each tide gauge location.

---

## Author Comment (AC1)

**Answer to review #1 of manuscript:**
**"Could old tide gauges help estimate past atmospheric variability?"**
**for *Climate of the Past**

Paul Platzer, Pierre Tandeo, Pierre Ailliot, Bertrand Chapron

March 2024

*Reviews in black,* *answers in blue,* **quotes from revised manuscript in bold.**

Summary: The study explores the possibility of using tide gauge records to reduce uncertainty in reconstructing past (19th century) sea-level pressure fields. The authors focus on the 20CR reanalysis, a long reanalysis product covering parts of the 19th century, and two tide gauge records in Northwestern France. The main conclusion is that the tide records can provide valuable information to constrain the 20CR reanalysis ensemble by attaching lower probabilities to the members of the ensemble that are less compatible with the implied sea level at those two tide records.

Recommendation: I think the idea of the study is interesting, and the study is, to a large extent, technically sound. I have some suggestions that the authors may want to consider in a revised version. My recommendation of 'major revisions' is more dictated by my interest in the revised version. The needed revisions are, in my opinion, between 'minor' and 'major'

We thank Reviewer #1 for the fruitful remarks and suggestions. The discussion has led to major improvements of the paper and we believe that the revised manuscript is more clear and precise.

Following also the remarks and suggestions of reviewer # 2, and in order to make the reviewed version of the paper more clear, we have made several modifications to the paper, which are listed below to make it clear to both reviewers.

i Pre-processing of the data has changed. The effect of mean-sea-level variations on surge was removed by removing a yearly median value of the surge instead of a time-constant + time-linear trend. The pressure used to assess the relationship with the sea-level is not the anomaly with respect to a given climatology anymore, but the difference between the local sea-level pressure at Brest and the sea-level pressure averaged over the whole North-Atlantic ocean.

ii The statistical relationship between sea-level and pressure is now estimated with a new model. First, this model is not linear, but local-linear. Is does not include the effects of pressure gradients (winds) anymore. This time we model pressure as a function of sea-level (the true objective of the paper), while in the original submission we modeled the sea-level as a function of the pressure and pressure gradients (winds). More details are given in section 3.1. of the revised manuscript.

iii As a consequence, section 3.2. was ruled out.

iv We kept only the tide gauge record of Brest, and we are not using the one of Saint-Nazaire anymore. In the way the algorithm was implemented and with our simple statistical relationship, using two tide gauges at a time was barely more informative than using just one. Therefore, for simplicity purposes we kept only the Brest record which is the longest one.

v For a better interpretation of applying our method in the 19th century, we have used independent pressure data for the city of Brest from the EMULATE project and recently release archive data from Météo-France. This data starts in 1855, while the Brest tide gauge starts in 1846. This data also greatly helps the interpretation of Figures 9, 12, 13 and 14 of the revised manuscript which shows specific events.

**General points**

**1.** This is somewhat a cosmetic comment, but the the manuscript refers to 'storm surges' whereas actually, the study considers the sea-level record after filtering out the tides and the long-term sea-level rise. In my understanding,

[Figure]

*Figure 4 in revised manuscript.* **Evaluation of the local-linear regression (LLR) on the period 1981-2015 using a leave-one-out procedure. (a) Histogram scatter-plot of the average estimate from the LLR versus true value of the MSLP difference with the reference ocean-averaged MSLP. (b) Density-histogram of normalized average error of the LLR estimate (black line) compared to theoretical probability density function of standard Gaussian random variable (dashed orange line).**

the residuals are not the 'storm surge' component. The storm surge is, by definition, caused by the passage of a storm. The mechanisms connecting the storm atmospheric forcing and the sea surface elevation are more complex in cases of storms than those for the 'normal' sea level. For instance, depending on the coastline topography, the wind may cause a direct elevation of sea-level in the direction of the wind. This is clear for estuaries, and it is the main cause of storm surges in many North Sea locations. The study statistically analyses the full range of sea-level variations, not only the extremes, and it is unclear why the text highlights 'storms'. This is only the cause for the last section, and this is related to some additional problems (see next point).

**1.** We have modified all occurrences of "storm surge" with simply "surge" to account for this comment.

**2.** The studies used a linear statistical regression method to link atmospheric predictors and sea-level elevation (after tidal and long-term trend filtering). The skill of the linear model is shown in Figure 4. It is visually clear that the linear model underestimates sea-level extremes, both high and low. This is very usual for linear models. But, while using tide gauges to constrain the reconstructed SLP would be permissible for 'normal' situations, it is clearly problematic for extremes. Thus, the last section needs a bit more attention, in my opinion. It could very well happen that the SLP from the ensemble members of the 20CR reanalysis passed through the linear model underestimates the storm surge (in this case, it is indeed a storm surge), and therefore, more ensemble members appear not compatible with the sea-level observations than they are. I suggest testing this approach with a recent storm when the 20CR has assimilated sufficient SLP observations to be considered accurate. In that recent storm, if I am correct in my concern, the study's approach would also find that the 20CR ensemble is biased towards implied lower storm surges. This would test the study's main claim, namely that 'storm surges' can constrain the 20CR reanalysis.

Even if the test results fall against the study's claim, the study itself can still be useful, albeit in a modified form. Long records would still be useful to constrain the 20CR, but this constraint in case of storms would be too restrictive, attaching too low a probability to too many ensemble members of the 20CR reanalysis. Alternatively, a non-linear model should be set up to more accurately estimate extreme-level from atmospheric forcing.

**2.** This is a very important remark and we thank reviewer #1 for pointing this out.

Indeed, using a linear model induces bias in the extremes. In order to partially remedy this problem, we have chosen to slightly change the relationship used in the study. Now, we use a local-linear regression (LLR) to predict the local MSLP in Brest from the observed surges. This was motivated by your remark, and by the fact that we observed a deviation from the linear relationship which depends on both the value of the 12h-averaged surge (see Fig. A) and on the value of the 12h-difference of the surge (see Fig. B).

As obvious from Fig. 4 of the revised manuscript (reproduced here), our local-linear model still induces biases in the extremes, although it is less biased for moderate values while the linear model was. However, we believe that this bias is less significant for our application purposes than the one induced by the ambiguity of the pressure reconstruction from surges due to the effects of wind. Looking at Fig. 9 of the revised manuscript (reproduced here), it is obvious that the surge-based LLR may actually either overestimate or underestimate the amplitude of pressure variations, depending on the amplitude and direction of the wind stress at the time of the storm. Therefore, we choose to still use a model that is biased in the extremes.

We expect the hypothesis of reviewer # 1 to be validated in case of typical (i.e. average) wind conditions: the

[Figure]

*Figure 9 in revised manuscript.* **Comparison of MSLP estimation in Brest from 20CRv3 (blue), LLR based on surges (orange), and independent observations (black dots) that were not used to build the orange and blue curves, for three periods surrounding the events studied in this section. Full lines correspond to average values while shaded areas correspond to ± one standard-deviation around the average.**

[Figure]

*Figure A. Not included in revised manuscript.* Quantiles $[0.\,, 0.05, 0.1\,, 0.15, 0.2\,, 0.25, 0.3\,, 0.35, 0.4\,, 0.45, 0.5\,, 0.55, 0.6\,, 0.65, 0.7\,, 0.75, 0.8\,, 0.85, 0.9\,, 0.95, 1.\,]$ of error of estimation from a simple linear regression of $\mathrm{MSLP}(t)$ based on 12h-averaged surge $h(t)$, as a function of $h(t)$, estimated by selecting the 200 closest values of $h(t)$ in the period 1981-2015. This figure is not part of the revised manuscript and is here only for discussion purposes.

LLR estimation of the most extreme pressure variations would be biased towards the mean value of MSLP, and our algorithm would select the members with the smallest MSLP deviation from the mean. We have not performed the test directly, but the discussion of section 5. of the revised manuscript sheds light on the limitations of our method line 368:

**Our claim that the wind variations are responsible for the persistent biases between the LLR pressure estimation and the reanalysis is supported by Fig. 13.f, where we also show the direction and amplitude of the 10m-wind intensity as given by the average over all reanalysis members and interpolated at the city of Brest. In March 1876, two low-pressure systems passed to the North of Brest's tide gauge, one around March 10th and a second around March 12th, as indicated by the reanalysis members and the independent pressure observations (Fig. 13.e). However, the first low-pressure system did not induce a surge as strong as the second one. One key difference between the two events is the wind amplitude, which reached 15m/s during the first event and then decreased to 5-10m/s during the second event, with almost steady wind direction. Although wind intensity and direction estimated from the reanalysis must be taken with care, the value of 15m/s is rarely exceed (only 7 in 1000 times in the period 1981-2015, not shown), indicating exceptional wind intensity during the event, and justifying the inaccuracy of the LLR which is based on already observed events and therefore biased towards typical wind conditions. Our interpretation relies on the fact that the effect of wind on extreme surges acts at small time scales (daily or sub-daily), which is backed by recent work (Pineau-Guillou et al., 2023).**

We emphasis these limitations but still believe that the study can be useful if improved, or that the methodology can be used for other variables than sea-level data.

**Minor comments** *(→ We added letters to allow cross-reference between reviews.)*

**A.** 'Anomalies are considered with a reference climatology computed as an average over all members and over $\pm30$ calendar days, $\pm 3$ day hours '.
This sentence is unclear.

This was modified in the revised manuscript. Moreover, since we are using a second type of preprocessing of MSLP, as suggested by reviewer # 2, we have added a whole subsection entitled "Preprocessing of mean-sea-level pressure" which is copied here (line 90)

**In this work, we are using only the mean sea-level pressure (MSLP) variable from 20CRv3. We make two different preprocessings of this variable.**

**A first preprocessing is used for the statistical relationship between the local pressure and the surge. As the latter is driven by a physical phenomenon called the "inverse barometer effect" which will be introduced in the next section, we consider the difference between the MSLP interpolated at the city of Brest (4.49504°W, 48.3829°N), and the MSLP averaged over all members of 20CRv3 and over the North-Atlantic ocean (using the reanalysis' land mask and averaging from 98°W to 12°E and from 0°N to 69°N), similarly to Ponte (1994). This spatial-averaged pressure is noted $\overline{\mathrm{MSLP}}^{ocean}(t)$ and depends only on time.**

[Figure]

*Figure B. Not included in revised manuscript.* Same as previous figure but as a function of the difference in surge $\Delta \overline{h}^{3h}(t)$ as defined in the revised manuscript. This figure is not part of the revised manuscript and is here only for discussion purposes.

**A second preprocessing of MSLP is used to compute the probability of transition from one member of the reanalysis to another in the Hidden Markov Model (HMM) presented in section 3.2. For this purpose, we consider seasonal anomalies of MSLP with respect to a climatology computed from the period 1847-1890, because the HMM is run only for those years. The reference MSLP climatology for calendar day $d$ and hour $h$ is given by the average over days between $d-30$ and $d+30$, hours between $h-3$ and $h+3$, and all years 1847-1890. This reference MSLP is noted $\overline{\text{MSLP}}^{clim}$ and depends on latitude and longitude.**

**B.** Table 1, Table 3 units are missing

These tables were removed as we do not use the linear regression anymore (see description above).

**References**

Lucia Pineau-Guillou, Jean-Marc Delouis, and Bertrand Chapron. Characteristics of storm surge events along the north-east atlantic coasts. *Journal of Geophysical Research: Oceans*, 128(4):e2022JC019493, 2023.

Rui M Ponte. Understanding the relation between wind-and pressure-driven sea level variability. *Journal of Geophysical Research: Oceans*, 99(C4):8033–8039, 1994.

---

## Author Comment (AC2)

**Answer to review #2 of manuscript:**
**"Could old tide gauges help estimate past atmospheric variability?"**
**for Climate of the Past**

Paul Platzer, Pierre Tandeo, Pierre Ailliot, Bertrand Chapron

April 2024

*Reviews in black, answers in blue, **quotes from revised manuscript in bold**.*

This work uses historical tide gauge records in the French coast as proxy data to improve the atmospheric 20th century reanalysis. Specifically, the sea level observations are used to constrain the number of ensemble members in the reanalysis, selecting and weighting them according to their correspondence to observed storm surges. This is an interesting idea. The approach discards model realisations that, although physically consistent with the scarce available mean sea level pressure observations, are inconsistent with observations of storminess from coastal tide gauges. I am not a statistician and therefore I am not qualified to comment on the details in section 4. My comments are focused on the results and, mostly, on sections 2 and 3 on data description and processing. I have found major issues in these sections, including a basic conceptual misunderstanding of the generation of storm surges. Also, there are missing details in data processing and I have concerns on the simple linear regression model that has been used to relate storm surges and sea level pressure anomalies.

We thank Reviewer #2 for the fruitful comments, corrections, remarks and suggestions. We believe that the review greatly improved the clarity and precision of the paper. More details are given below.

In the revised manuscript, we have chosen to do some major modifications, driven by the comments of both reviewers.

i Comments of reviewer # 2 on the physical mechanisms driving storm surge generation have been taken into account and important modifications were made both in the manuscript's text and in the data pre-processing. The effect of mean-sea-level is now removed using yearly medians. A reference pressure averaged over the ocean is used to estimate the statistical relationship between sea-level and local pressure.

ii The linear model was modified for a local-linear model, details of which are given in section 3.1 of the revised manuscript. However, we want to stress here that the change in the model only mildly affected the result of our algorithm. However, we believe that the revised manuscript makes a clearer interpretation of the results and of the model's advantages and limitations, thanks to the added value of the reviewer's comments suggestions.

Another major concern is that the authors state in the conclusions that they use the tide gauge time series as a barometer record not assimilated in the reanalysis. However, they model the storm surges from sea level accounting for both pressure and geostrophic winds. I do not understand the interest of including the winds here (which by the way are not modelled correctly, in my opinion). If the purpose is to incorporate only pressure-like measurements, then pressure gradients should not be included at all. Or at least, the pressure-like record should be reconstructed without accounting for wind effects. All these issues are detailed in the following. My recommendation is that the manuscript should undergo major revisions before being considered for publication.

Accounting for this comment, we have chosen to remove the part of the statistical model related to pressure gradients. In our modified statistical model, we estimate directly the probability distribution of pressure based on the tide gauge record. More details are given in the article and copied below.

Other major modifications to the manuscript have been made, which are listed here for clarity purposes:

i We have removed the use of the Saint-Nazaire tide gauge, and kept only the Brest tide gauge record. In the way our statistical model was built, adding the Saint-Nazaire tide gauge observation was not making a significant change to the results, as in most cases the response of the Saint-Nazaire tide gauge to variations of pressure was nearly identical to the one of Brest. Therefore, adding it only complicated the paper's statement. Only a more sophisticated model would have been able to make smart use of the slight differences between both sea-level responses, and this is out of the scope of this article.

ii We have used independent pressure records for the city of Brest, coming from the EMULATE project and from recently released Météo France archive data. This motivated us further both to keep only the Brest tide-gauge and not to incorporate pressure-gradients in the statistical model. This new set of observations is described in section 2.5 of the revised manuscript. In greatly helps to identify the potential and the limitations of our method, as explained below and in section 5 of the revised manuscript.

**Section 2:** This section needs further details in the description of the data set. Preprocessing of data needs improvement.

**1.** L.62: define the spatial and temporal resolution of the reanalysis.

**1.** We have added the following sentences line 71:

**It has a temporal resolution of 3 hours, and uses a spectral triangular model in space with truncation of T254 (approximately 75km at the equator). There are 64 vertical levels, up to .3mb.**

**2.** L. 70: I assume the variable used is mean sea level pressure (MSLP), correct?

**2.** Correct. In section 2.2 of the revised article called "Preprocessing of mean-sea-level pressure", at the beginning we state 90:

**In this work, we are using only the mean sea-level pressure (MSLP) variable from 20CRv3.**

This whole new section is copied below in answer to comment **4.**

**3.** L. 81-82: please, remove.

**3.** We have removed the following sentence: "This double constraint of data scarcity and high variations in sampling frequency in the 19th century legitimizes the search for other sources of observations to study centuries-old atmospheric variability."

**4.** L. 83: anomalies of what? I assume this means that mean SLP anomalies are computed with respect to climatologies for 2 periods. Please, specify.

**4.** This has changed slightly (anomalies are not considered for the satellite-era period anymore). This is specified more precisely in the new subsection 2.2 called "Preprocessing of mean-sea-level pressure", which states line 90:

**In this work, we are using only the mean sea-level pressure (MSLP) variable from 20CRv3. We make two different preprocessings of this variable.**

**A first preprocessing is used for the statistical relationship between the local pressure and the surge. As the latter is driven by a physical phenomenon called the "inverse barometer effect" which will be introduced in the next section, we consider the difference between the MSLP interpolated at the city of Brest (4.49504°W, 48.3829°N), and the MSLP averaged over all members of 20CRv3 and over the North-Atlantic ocean (using the reanalysis' land mask and averaging from 98°W to 12°E and from 0°N to 69°N), similarly to Ponte (1994). This spatial-averaged pressure is noted $\overline{\mathrm{MSLP}}^{ocean}(t)$ and depends only on time.**

**A second preprocessing of MSLP is used to compute the probability of transition from one member of the reanalysis to another in the Hidden Markov Model (HMM) presented in section 3.1. For this purpose, we consider seasonal anomalies of MSLP with respect to a climatology computed from the period 1847-1890, because the HMM is run only for those years. The reference MSLP climatology for calendar day $d$ and hour $h$ is given by the average over days between $d-30$ and $d+30$, hours between $h-3$ and $h+3$, and all years 1847-1890. This reference MSLP is noted $\overline{\mathrm{MSLP}}^{clim}$ and depends on latitude and longitude.**

**5.** Description of tide gauges (section 2.2): please, define the temporal sampling of tide gauges, and their overlapping periods. The statement that one time series can be used to fill in gaps of the another is incorrect. They are highly correlated at low frequencies but not necessarily at high frequencies (i.e. storms). Actually, they are not even in the same location, as stated later in the text, so differences in storm surges may be expected.

**5.** The overlapping periods need not be specified anymore since the Saint-Nazaire tide gauge is not used anymore. The temporal sampling was specified in the following sentence line 106:

**The Brest sea-level record from this database starts in 1846 and has a hourly sampling.**

Indeed, it is incorrect to state that one series can be used to fill the gaps of the other. This is only partially true for 12h-averages of sea-level, of which the tide was removed as well as the yearly median. Anyway, we do not use the Saint-Nazaire tide gauge anymore, so the statement was removed in the revised article.

**6.** L. 97: actual → modern, current, present-day?

**6.** Indeed. The corrected sentence is now line 108:

**This combination of historical and modern records is at the foundation of the methodology exposed in the next section.**

**7.** The removal of MSL effects in the tide gauge processing is incorrect. The impact of MSL should not be removed in this way, because there are other MSL variations at interannual and decadal time scales that would

[Figure]

*Figure 3 of revised manuscript, reproduced here.* **An example output of the different stages of preprocessing of the sea-level signal used in this work. (a) Raw level before (full, blue line) and after (dashed, orange line) removing the tidal part of the signal. (b) Sea-level after removing the yearly median value: the surge $h(t)$ (1h sampling, orange dashed curve), the centered 12h-average of the surge $\overline{h}^{12h}(t)$ (green full curve), and the 12h difference between 3h-averages of the surge $\Delta\overline{h}^{3h}(t)$ (gray dotted curve).**

remain. A better approach is to remove mean (or median) yearly averages. It also avoids the problem of selecting an arbitrary change point to calculate different linear trends.

**7.** This was changed in the revised manuscript. We are now removing the yearly median after having removed the tidal part of the signal, as detailed in section 2.4 of the revised manuscript. We copy-paste this section below (starting line 111):

**As mentioned earlier, the part of the sea-level which responds to atmospheric processes is the surge (also called "storm surge" or "skew surge"). To access the surge, one first has to remove the tidal part of the signal, and then to remove yearly variations of the mean-sea-level (at interannual and decadal scale), such as sea-level rise (Cazenave and Llovel, 2010). In this work, we are also interested in moving averages and differences of the surge. All these steps are exemplified in Fig. 3.**

**We first compute the tidal constituents of the raw sea-level (blue curve, Fig. 3.a) using U-Tide (Codiga, 2011), which performs harmonic (Fourier) decomposition with prescribed frequencies corresponding to planetary movements. The tidal constituents are computed over two different periods, one is 1847-1890, and the second is 1981-2015. Removing the tidal part of the signal gives the orange dashed line of Fig. 3.a, which has a temporal average value of $\sim$4m for the Brest tide gauge.**

**Then, we remove the yearly median value of the sea-level, which allows to access the surge (orange dashed line of Fig. 3.b). We choose to remove the median and not the mean because the mean can in principle be influenced by the number and magnitude of extremes in a given year, which can be linked to the number and magnitude of storms passing in a given year. This second step allows to access the surge which is noted $h(t)$ in the following:**

$$h(t) = H(t) - \mathrm{Tide}_H(t) - \mathrm{median}\left[H(t'),\, t' \in \mathrm{year}(t)\right] \ ,$$

**where $H(t)$ denotes the raw sea-level, $\mathrm{Tide}_H(t)$ is the tidal part of the signal computed from $H$, and year$(t)$ is the year in which time $t$ is found.**

**Note from Fig. 3.b that the surge fluctuates at hourly scale, part of which are oscillations which are not due to variations in atmospheric pressure. These oscillations are either due to tide-surge interactions (Horsburgh and Wilson, 2007) or to measurement errors in the 19th century leading**

to phase shifts. Such oscillations can dominate the surge signal in Brest where the tidal amplitude is large. Furthermore, tide-surge interactions lead to stronger surges in low-tide and weaker surges in high-tide (Horsburgh and Wilson, 2007). As these phenomena are not linked to atmospheric processes, we chose to filter them out with a simple 12h-average (green full curve in Fig. 3). This also implies that these 12-hours-averaged surges will only respond to atmospheric events persisting for more than 12 hours. Given the spatial resolution of 20CRv3, smaller-scale events are likely not to be represented in the MSLP fields used in this study. In the following, we note $\overline{h}^{12h}(t)$ the 12h-average of the surge:

$$\overline{h}^{12h}(t) = \frac{1}{12} \sum_{t'=-6}^{t'=+6} h(t+t') \,.$$

Furthermore, as we are using sea-level observations to estimate atmospheric pressure, we also want to measure the amplitude of local time-variations of the surge. Indeed, as will be further explained in section 3.1, the sea-level response to variations of pressure depends on the time-scale of these variations. More precisely, the "inverse barometer" is an approximation that is only valid for slow variations of pressure. Accordingly, when observing fast variations of the surge, one expects deviations from the inverse barometer approximation. We therefore compute the difference between the surge at time $t$ and at time $t - 12h$, choosing the 12h-interval again to filter out oscillations at a period close to 12h. Furthermore, since the reanalysis is run at 3h-resolution, we perform a 3h-moving average of the surge before computing the difference. This difference is noted $\Delta \overline{h}^{3h}(t)$ and defined by the following equation:

$$\Delta \overline{h}^{3h}(t) := \frac{1}{3} \sum_{t'=-2}^{t'=+1} \left[ h(t+t') - h(t-12+t') \right] \,.$$

**8.** L. 108-109 on detiding: which is the period used for detiding?

**8.** We specify this in the revised manuscript, see above and section 2.4 of the revised manuscript. We recall here line 116:

**The tidal constituents are computed over two different periods, one is 1847-1890, and the second is 1981-2015.**

The first period is used to remove the tidal signal in 1847-1890 while the second is used to remove the tidal signal in 1981-2015.

**9.** L. 113: phenomenons→ phenomena

**9.** This was corrected in the revised manuscript.

**10.** L. 110: A 12-h filtering removes part of the storm surge variability, especially that related to storms.

**10.** Indeed. We have not found an other simple method that allows to suppress the oscillations unrelated to atmospheric variations from the tide gauge signal.

**11.** L. 114-115: smaller scale events are unlikely to be recorded in 20cR at 2deg resolution

**11.** This is true and we should have noted it from the start. This argument is now included in the revised manuscript as (line 132):

**Given the spatial resolution of 20CRv3, smaller-scale events are likely not to be represented in the MSLP fields used in this study.**

**Section 3:** This section contains fundamental errors in the understanding of the processes that generate storm surges. Also, the linear regression model is questionable as applied here, and should be better justified. More details:

As stated above, we have made several modifications to the MSLP pretreatment and to the statistical model, including an ocean-averaged reference pressure, discarding pressure gradients, and using a local-linear model. More details are given below.

**12.** L. 123-124: hydrostratic equilibrium is a general law not only affecting this region. Inverted barometer has been widely applied everywhere.

**12.** We believe that the new formulation does not imply that the inverse barometer would be an effect only valid in Brest. See line 162:

**The filtered surges described in section 2.4 respond to sub-seasonal variations in atmospheric pressure. First, the sea-level is sensitive to pressure variations. An approximation called the "in-verse barometer effect" (Roden and Rossby, 1999) states that an increase (respectively, decrease) of 1hPa in pressure at the mean sea-level leads to a decrease (respectively, increase) in sea-level of ap-proximately 1cm. This approximation is valid for slow variations of atmospheric pressure compared to the typical time of dynamic adjustment of the sea-level (Bertin, 2016).**

[Figure]

1846 ISPDv4.7 Number of Observations/Day

NOAA/PSL & CU/CIRES

*Figure A. Not part of revised manuscript.* Number of observations per day assimilated in 20CRv3 for year 1846.

**13.** L. 127-131: this is incorrect. The main mechanism of wind as storm surge generation is not the Ekman transport, but the piling up of waters due to wind blowing perpendicular to the coast. Check your own results, where you can actually see it (e.g. line 163). Please, change this in the abstract too.

**13.** This was modified in the text, see below. The abstract does not mention directly the effects of wind anymore as we do not use it in our statistical relationship. See line 167:

**Moreover, the piling up of waters due to wind blowing perpendicular to the coast is responsible for positive (respectively, negative) surge when the wind stress is directed towards (respectively, away from) the coast. This effect depends non-linearly on the wind amplitude (Bryant and Akbar, 2016; Pineau-Guillou et al., 2018). Statistical correlation between pressure variations and wind intensity and direction are responsible for deviations from the inverse barometer approximation (Ponte, 1994).**

**14.** L. 1334-134: I assume that SLP here (and used later in equation 1) is the SLP anomaly with respect to time at the given grid point. If this intend to represent the inverted barometer effect, this is incorrect. The anomaly should be computed with respect to the average mean sea level pressure over the world oceans in the reanalysis. See for example Ponte 1994 (https://doi.org/10.1029/94JC00217).

**14.** Our use of MSLP anomaly with respect to seasonality was incorrect. We now use the average over the North-Atlantic ocean -as in Ponte (1994)- from the reanalysis as a reference pressure. We have also tried to use the average over the world oceans (not only the North-Atlantic), and it only slightly changes the performance of the statistical method, so we decided to keep it as is. Using the North-Atlantic ocean average may be more meaningful in the 19th century when the distribution of observations is not uniform, see Fig. A of this document (this figure does not appear in the revised manuscript and is only here for discussion purposes).

Note that we still use the climatological anomaly to compute the probability of transition from one member to another in the HMM algorithm. This is all described in section "Preprocessing of mean-sea-level pressure", line 92:

**A first preprocessing is used for the statistical relationship between the local pressure and the surge. As the latter is driven by a physical phenomenon called the "inverse barometer effect" which will be introduced in the next section, we consider the difference between the MSLP interpolated at the city of Brest (4.49504°W, 48.3829°N), and the MSLP averaged over all members of 20CRv3 and over the North-Atlantic ocean (using the reanalysis' land mask and averaging from 98°W to 12°E and from 0°N to 69°N), similarly to Ponte (1994). This spatial-averaged pressure is noted $\overline{\mathrm{MSLP}}^{ocean}(t)$ and depends only on time.**

**A second preprocessing of MSLP is used to compute the probability of transition from one member of the reanalysis to another in the Hidden Markov Model (HMM) presented in section 3.2. For this purpose, we consider seasonal anomalies of MSLP with respect to a climatology computed from the period 1847-1890, because the HMM is run only for those years. The reference MSLP climatology**

**for calendar day $d$ and hour $h$ is given by the average over days between $d - 30$ and $d + 30$, hours between $h - 3$ and $h + 3$, and all years 1847-1890. This reference MSLP is noted $\overline{\mathrm{MSLP}}^{clim}$ and depends on latitude and longitude.**

**15.** L. 134: interpolated how? Linearly?

**15.** Yes, we use linearly interpolated MSLP at the city of Brest. This was made clear in modified section 3.1 "Local Linear Regression (LLR) between surges and mean-sea-level pressure" line 188:

**The predicted variable is $\mathrm{MSLP}(t) - \overline{\mathrm{MSLP}}^{ocean}(t)$ where $\mathrm{MSLP}(t)$ is the value of the MSLP linearly interpolated at the city of Brest from the reanalysis.**

**16.** Equation 1: please, define all the terms. I do not believe this is a good way to separate the forcings. I think it would be better to use perpendicular and parallel to the coast MSLP gradients (geostrophic winds). In this case, the parallel will be insignificant and the model would be simplified. This separation has a physical meaning and would also allow to interpret the results easily.

**16.** This equation was suppressed as we do not use the MSLP gradients anymore in the statistical relationship, as suggested by reviewer # 2.

**17.** Also, why is the term Cov introduced? How does it change the results?

**17.** This term was introduced because there are high correlations between the observed surges in Brest and Saint-Nazaire, especially after 12h-filtering. This term is removed now since we use only Brest's tide gauge.

**18.** Still in the same equation, are all coefficients significant? what happens if a step-wise regression is used instead? I would expect gamma coefficients to be discarded.

**18.** We are now using a local-linear relationship where all coefficients are re-estimated for each new value of the regressors, which are now the surge's 12h-average and 12h-difference. More details on the model are given in section 3.1 "Local Linear Regression (LLR) between surges and mean-sea-level pressure".

**19.** Tables 1 and 2: units are missing.

**19.** These tables were removed in the revised version of the manuscript.

**20.** L. 170: "1.02 on physical grounds" → actually not, the inverted barometer effect is a simplification and does not work at the coast.

**20.** This line is not present in the revised version of the manuscript.

**21.** L. 171-172: "This justifies..." → I do not think this statement is correct. I believe a step- wise regression would tell you if the use of these parameters is justified or not. Here it has not been proved, not even with uncertainties of the parameters.

**21.** This line is not present in the revised version of the manuscript. The new method is not a global linear-regression, and therefore step-wise regression cannot be used to justify the choice of parameters. We have chosen to justify our model by other means, see below.

**22.** L. 174: wave setup would be removed with the 12-h averaging, so it does not play any role.

**22.** Indeed our understanding of the physical mechanism called "wave setup" was not correct. This line is not present in the revised version of the manuscript.

**23.** Section 3.2: To demonstrate that the same coefficients can be used during another period, this is not the correct approach. I think a better approach would be to use the recent period to generate 19th-century-like observations, i.e., downgrading the number and location of observations, and compute the coefficients of the LR. These downgraded, coarse-resolution coefficients would be directly comparable to the ones in the section above with the full resolution. It would be important to calculate uncertainties in all cases in order to compare the coefficients.

**23.** The methodology proposed by reviewer #2 to demonstrate that the coefficients can be used in both periods is interesting, and would definitely be the best way to prove that the statistical relationship derived in recent years also holds in previous years. However, if our understanding of what you suggest is correct, it is not feasible for us in practice. Indeed, we neither have access to the numerical weather prediction model, nor to the data assimilation scheme used to generate 20CRv3, and we are not in capacity to re-run the reanalysis with downgraded number of observations. This is beyond our capabilities and out of the scope of our study.

This is the reason why we chose to include independent pressure observations, as stated earlier. This provides an independent data to compare to both 20CRv3 and our statistical estimate of pressure based on surges. Aside from the fact that it is a more robust and convincing justification, it also allows to better understand the limitations of our methodology. More details are given in the manuscript and below.

For these reasons, and because it would not be feasible with our new, local-linear regression, we have chosen not to replicate this section.

**24.** Fig 5: it is unclear what is shown here. Are coefficients of the modern period used here? Please, specify. The description of the results in fig 5 is confusing and does not reflect, in my opinion, the results. For example, in l. 188 the bias referred to one of the cases is visible in all (5a and 5b); l. 193: which interpretation is referred to here?; l. 196 states that there is no clear sign of bias in individual members but I think the bias is similar to that in the ensemble average. The bias is not only due to ensemble mean but also to limitation of coarse resolution data.

**24.** This Figure was removed from the final manuscript because the method was changed as stated earlier.

**25.** L. 206-220: this discussion on the differences in coefficients should consider uncertainties to ensure that the use of two different periods lead to different values. Values slightly larger/smaller do not provide confidence in the results. Therefore, the interpretations are not reliable (e.g. differences attributed to ensemble averaging).

**25.** Again, this discussion was removed.

**26.** L. 225-226: I think the consistency of the coefficients and the explained variance in modern and old periods is a consequence of the dominance of the inverted barometer effect. This is probably not true for extreme values generated by strong winds associated to storms, but it holds for mean storm surge variability driven by pressure changes.

**26.** This comment also motivated us to remove the pressure-gradient part of our algorithm. We believe that our new algorithm, although still mostly imperfect, is able to capture part of these two regimes: the moderate values where the inverse barometer is more valid, and the extreme values where the effect of wind causes the statistical relationship to differ from the inverse barometer. However, our algorithm is still biased towards already observed values of pressure, and therefore has a hard time modelling the largest absolute values of pressure, as explained further in section "Local-linear regression (LLR) between surges and mean-sea-level pressure". However, we believe that it is still a better model than the previous linear one.

**27.** Summarising, I believe the LR model should be modified to consider winds parallel and normal to the coast in each case, and use step-wise (or similar) approach to remove the terms that do not explain more variance but introduce noise. Also, uncertainties of the parameters should be calculated. The use of the coefficients calculated in the modern period should be tested in downgraded modern data to prove that they are usable in older periods. The entire comparison and discussion of the two periods should be modified accordingly and simplified in case this is proved.

**27.** In conclusion, we have chosen to follow the alternative path suggested by reviewer #2, that is to discard the use of MSLP gradients in our model. We do not calculate the uncertainties in the parameter estimation of our new model because the parameters change at each time $t$. Rather, we chose to evaluate the consistency of the *result* of our algorithm, that is, the value of the predicted average $m(t)$ and variance $\text{var}(t)$ of the distribution of $\text{MSLP}(t) - \overline{\text{MSLP}}^{ocean}(t)$, as defined in the text. To check the consistency of the result of our algorithm with the ground truth in the period 1980-2015, we compared $m(t)$ estimated from our algorithm with the true value $\text{MSLP}(t) - \overline{\text{MSLP}}^{ocean}(t)$ in a density-scatter plot (Fig. 4.a of the revised manuscript), and to assess the consistency of our estimate of variance we checked that the variable $\text{MSLP}(t) - \overline{\text{MSLP}}^{ocean}(t)$ shifted by $m(t)$ and rescaled by $\text{var}(t)^{1/2}$ follows a standard Gaussian distribution using a histogram compared to a standard normal probability distribution function (Fig. 4.b of the revised manuscript). These tests demonstrate the relevance of the model when applied in the period 1980-2015.

To assess the relevance of our algorithm in the period 1848-1890, we can use the independent observations of pressure, after making a yearly-shift to avoid constant differences in pressure as an observational artifact (see section 2.5 of the revised manuscript). These pressure observations are compared to our estimate using surge observations, and to the estimate based on the reanalysis. Several examples are shown in section 5. (figures 11, 12, 13, 14), with one example (figure 13) where we also show the time series of 10m-wind as estimated from 20CRv3, because it helps interpreting the erroneous estimation of pressure from our LLR algorithm for a specifi storm. The main conclusion of section 5. is that wind is the key driver of uncertainties in our LLR-based estimation of pressure (one value of surge cannot be unambiguously attributed to either pressure or wind, at least in the way we modelled it). We give some more details here that answer part of reviewer #2's questions.

In Fig. B of the present document, we show the statistical relationship between the daily 10m-wind direction as estimated from the reanalysis versus the daily value of surge. Projecting the wind component along the direction $\pi/4$ shows a non-trivial relationship between wind and surges, as shown in Fig C of the present document. These figures are here for discussion purposes and are not part of the revised manuscript.

In the revised manuscript, Fig. 13 shows that the same low value of MSLP (as estimated both by the 20CRv3 reanalysis and the independent pressure observations for the city of Brest) can lead to different values of surge, and we interpret this as the result of a decrease in the amplitude of wind from an exceptionally high value ($¿15$m/s) to a moderate value. When the wind intensity is exceptionally high, the estimate of pressure from the surge-based LLR is biased, while it is not when the wind intensity decreases to more moderate, typical values. We believe that this is the main limitation of our study. However, the study can still be useful if improved, and our combination of HMM and statistical relationship could be used to assess other reanalysis products from other independent observations.

**Section 4:** I am not an expert in HMM and I am not qualified to comment on the details of the method explained here. It would be good that an expert statistician review s this part. I nevertheless would recommend defining all variables in the equations, as well as the acronyms (e.g. NWP), as it makes it very difficult to follow as it is now. Other comments follow:

**28.** L. 270: this is a rather large area. Are the results sensitive to this choice?

**28.** Yes, the results are, in principle, sensitive to the choice of this area, as it is used to compute the probability of transition from one member to another. However, as MSLP fields are highly space-correlated, the choice of the

[Figure]

*Figure B. Not part of revised manuscript.* The effect of wind direction on the daily-average of surge, using daily 10m-wind data linearly interpolated at the city of Brest from the ensemble mean of 20CRv3, for the period 1981-2015. This shows a preferred direction at $\sim \pi/4$ for positive surges and opposite direction $\sim -3\pi/4$ for negative surges.

[Figure]

*Figure C. Not part of revised manuscript.* Same as Fig. B but for projection of wind on the direction $\pi/4$.

extent of the area is of lesser influence than the choice of the parameter $\theta$, which we set to a value which allows to maximize the likelihood of the reanalysis based on the surge observations. In particular, high values of $\theta$ correspond to high probabilities of transition from one member to another, and low values of $\theta$ correspond to low probabilities of transition. Whatever the choice of area, in the limit of $\theta \to 0$ transition from one member to another are impossible, and in the limit $\theta \to +\infty$ all transitions between members are allowed. Therefore, once the area is chosen, the critical parameter is $\theta$.

**29.** L. 336-337: does this mean that the storm surges constrain the data more in winter when they have a stronger signal?

Yes this is correct.

**30.** L. 361: then→than

The sentence in the revised manuscript is unchanged "Note, as well, that the area of influence is greater for $\delta\mu_{\text{HMM}}$ then for $\delta\mu_{\text{HMM}}$, because of the time-propagation of corrections thanks to the smoothing HMM algorithm." We do not understand why "then" should be replaced by "that" here.

**31.** Section 4.3 is very illustrative of the potential of the approach. However, more information would be useful to understand why including a single (or two) new records essentially rules out 79 out of 80 model ensemble members. In particular, how many sea level pressure records were included in this period (or shortly before the storms) in the area? How are they distributed? It would be also useful to see a case with more sea level pressure observations (early or mid- 20th century for example).

Fig. 10 of the revised manuscript shows the distribution in space of the number of observations assimilated in the reanalysis during the months of the studied events.

We have included a case in year 1888 where the reanalysis is more constrained by pressure observations. The four time series in Fig. 9 clearly show how the reanalysis is more and more constrained over time and therefore has reduced uncertainties. The case of Fig. 14 (in year 1888) also shows higher values of the effective number of members compared to the other cases.

**Conclusions section:**

**32.** L. 405-408: I understood from the LR model that winds were incorporated into the model as SLP gradients. Then, why if wind is an important driver of storm surges may limit the use of the tide gauge? In fact, if wind is not accounted for and wants to be removed, then the LR model could include only the inverted barometer effect with the adjusted parameter at each tide gauge location.

**32.** The proposed approach is the one we have chosen in the revised manuscript. The revised conclusion now reads line 405:

**This study is a proof of concept for the use of century-old tide gauge data as a means of understanding past atmospheric subseasonal variability. Surges of Brest allow to assess part of the atmospheric variability that was uncaught in global 20CR reanalyses based on pressure observations. Weighing 20CR members according to surge observations reduces the effective ensemble size, and implies significant deviations in members-averaged sea-level pressure in the Bay of Biscay. Through the second-half of the 19th century, these deviations diminish and the effective ensemble size rises, however they remain non-negligible. Independent pressure observations in the city of Brest are coherent with pressure estimations from the reanalysis and the surge-based local-linear relationship. Such comparisons also show that the reconstruction of pressure based on surges is ambiguous due to the influence of winds, so that biases between the surge-base and the reanalysis-based pressure estimates can last for several days.**

**This work has several potential applications. First, replicating this work with other tide gauges could help to validate reanalyses like 20CRv3 against independent data, and to potentially identify anomalous trends or wrong estimation of specific events. Combining our statistical approach with the physics-based approach of Hawkins et al. (2023) could allow to have both a precise estimate from a high-fidelity coastal model and a good quantification of uncertainties. Second, tide gauges could be used to constrain regional scale atmospheric simulations in order to better estimate the magnitude and spatial extent of known past severe storms. Third, tide gauge records could be combined with direct observations of atmospheric pressure to give statistical estimates of atmospheric fluctuations in the 19th century without the use of a Numerical Weather Prediction model, such as the optimal interpolation of Ansell et al. (2006) based on direct pressure observations only, or the analogue upscaling of Yiou et al. (2014) for the short period 1781-1785 of dense observations in western Europe. Finally, this work could be replicated in a more general context, using other types of variables and observations, learning the relationship between observations and large-scale features using recent observations and precise reanalyses, and applying these statistical relationship in the past to uncover past large-scale events. In particular, the hidden-Markov model algorithm outlined here could be replicated to weigh ensemble members according to independent observations.**

**References**

Tara J Ansell, Phil D Jones, Rob J Allan, David Lister, David E Parker, M Brunet, Anders Moberg, Jucundus Jacobeit, P Brohan, NA Rayner, et al. Daily mean sea level pressure reconstructions for the european–north atlantic region for the period 1850–2003. Journal of Climate, 19(12):2717–2742, 2006.

Xavier Bertin. Storm surges and coastal flooding: status and challenges. La Houille Blanche, (2):64–70, 2016.

Kyra M Bryant and Muhammad Akbar. An exploration of wind stress calculation techniques in hurricane storm surge modeling. Journal of Marine Science and Engineering, 4(3):58, 2016.

Anny Cazenave and William Llovel. Contemporary sea level rise. Annual review of marine science, 2:145–173, 2010.

Daniel L Codiga. Unified tidal analysis and prediction using the utide matlab functions. 2011.

Ed Hawkins, Philip Brohan, Samantha N Burgess, Stephen Burt, Gilbert P Compo, Suzanne L Gray, Ivan D Haigh, Hans Hersbach, Kiki Kuijjer, Oscar Martínez-Alvarado, et al. Rescuing historical weather observations improves quantification of severe windstorm risks. Natural hazards and earth system sciences, 23(4):1465–1482, 2023.

KJ Horsburgh and C Wilson. Tide-surge interaction and its role in the distribution of surge residuals in the north sea. Journal of Geophysical Research: Oceans, 112(C8), 2007.

Lucia Pineau-Guillou, Fabrice Ardhuin, Marie-Noëlle Bouin, Jean-Luc Redelsperger, Bertrand Chapron, Jean-Raymond Bidlot, and Yves Quilfen. Strong winds in a coupled wave–atmosphere model during a north atlantic storm event: Evaluation against observations. Quarterly Journal of the Royal Meteorological Society, 144(711): 317–332, 2018.

Rui M Ponte. Understanding the relation between wind-and pressure-driven sea level variability. Journal of Geophysical Research: Oceans, 99(C4):8033–8039, 1994.

Gunnar I Roden and H Thomas Rossby. Early swedish contribution to oceanography: Nils gissler (1715–71) and the inverted barometer effect. Bulletin of the American Meteorological Society, 80(4):675–682, 1999.

P Yiou, M Boichu, R Vautard, M Vrac, S Jourdain, E Garnier, F Fluteau, and L Menut. Ensemble meteorological reconstruction using circulation analogues of 1781–1785. Climate of the Past, 10(2):797–809, 2014.

---

## Referee Report (RR1)

The authors have made substantial changes to the manuscript in response to my previous concerns. I still believe that this idea is original and worth exploring. Again, I am not commenting on the statistical model, as it is beyond my expertise. The results though seem to make sense at constraining the ensemble members in 20CR dataset.

Despite the undoubtful interest of the work, I still have a number of comments on the manuscript, some minor, some other that would require further exploration. I am listing all of them below, as they appear in the text. Many of my comments arise from wrong statements or parts of the text that are unclear. This makes the manuscript sometimes hard to follow, even in the descriptive parts.

My first comment is about the format of the author tracked changes document. Please, avoid this practice in future submissions. It makes impossible for the reviewer to go through this file and identify the changes and it makes reading very uncomfortable. This is not what I understand should be a version with tracked changes which should contain a comparison between the original and the revised versions. I have therefore gone through the new file, without paying much attention to the tracked version. All the lines below thus refer to the new version.

- The term surge is misleading. Please, use the terminology as defined in the literature to avoid misunderstandings: https://link.springer.com/article/10.1007/s10712-019-09525-z.

- Abstract: the storm surge is not the response to atmospheric pressure. It also includes winds and waves and mean sea level.

 - l. 72: .3 mb should be 0.3 mbar (I guess)

- l. 93 "As the latter is driven by a physical phenomenon called the "inverse barometer effect". This statement is incorrect, see my comment and reference above.

- l. 92-97. This is probably a good approximation but it is not strictly correct. First, the anomaly of every ensemble member should be calculated with its own mean. Second, the average should be calculated within the global ocean. According to the responses to my previous revision, the latter has been checked and made negligible differences. You should need to check the first point though.

- please remove figure 2. It does not make much sense to have only one station. This can be mentioned in the text.

- l. 112: storm surge and skew surge are two different metrics

- l. 112: "To access the surge, one first has to remove the tidal part of the signal, and then to remove yearly variations of the mean-sea-level (at interannual and

decadal scale), such as sea-level rise". This is incorrect. The storm surge is the difference between sea level and tides, so it does include mean sea level, so the definition is not exact. Then mean sea level can be removed if the purpose is to understand short term changes.

- figure 3: in the legend and caption, level should be sea level

- l. 127: "These oscillations are either due to tide-surge interactions (Horsburgh and Wilson, 2007) or to measurement errors in the 19th century leading to phase shifts" So you mean that these are 12-h oscillations? it is hard to see in the figure.

- l. 131: "This also implies that these 12-hours-averaged surges will only respond to atmospheric events persisting for more than 12 hours." I do not think this is correct. It will smooth higher frequency changes but not remove them completely.

- l. 135-142: the inverted barometer acts at periods shorter than 12h, so the statement about longer periods is incorrect. Also, what do you mean by "local time-variations"? In any case, because this record is to be compared to a very low resolution model output, this approach may not be critical. I don't think it is applicable to high resolution models though.

- l. 151-156: this is not clear to me. I understand that pressure observations have biases. Are you correcting for yearly dependent biases? If so, this is altering the temporal variability of the observations. This means that they cannot be compared as independent from the model. Bias-corrections must apply the same bias during the entire record. If the record is made of two separate set of observations then it makes sense to apply two biases separately, but never every year.

- l. 169-171: "Statistical correlation between pressure variations and wind intensity and direction are responsible for deviations from the inverse barometer approximation of the statistical linear relationship between surges and pressure (Ponte, 1994)" I am unsure about what this means. What statistical correlation?

And after this: "As a consequence of these combined effects of wind and pressure, the statistical relationship between the filtered surges and the pressures from 20CRv3 is expected to be non-linear, and not deterministic. " Again, unclear. I would say that the effect is linear but there are other processes.

Next: "As showed by Hawkins et al. (2023), using a physical coastal model forced by the values of pressure (and winds) from the 20CR can lead to biases in the estimation of associated surges due to the resolution of the reanalysis, so that a statistical model is needed to correctly represent uncertainties" The need of a statistical model is not really an implication of the inability of the coarse resolution numerical models.

- L. 211-215 on the validation of the LLR model. If I understood correctly, the model is applied to a part of the dataset for the period 1980-2015 (those for which neighbours are 14 days apart). Then, Figure 4 should represent the comparison between the true and the modelled values only for those part of the time series that were not used to fit the model. Is this what it is showing? From the caption and text it seems that it is showing all values, but in this case, it is not a validation because values used to fit the model are also in there. Please, clarify.

- How are the results of the LRR different from purely inverted barometer approximation? Is Figure 4 improved with the LRR with respect to the most simple approach? In principle, if only subseasonal pressure variations are targeted, inverted barometer is likely a good proxy for sea level changes. If another more complex model is to be used, its benefits should be demonstrated. I am afraid that at this point it is still unclear to me why this model is needed.

-l. 358-359: "In 1865 (Fig. 9.b), although the surge-based reconstruction happens to be more consistent with observations than the reanalysis, the reverse is also true." Please, rephrase, two opposite things cannot be true …

-l. 362: "We attribute these biases to different atmospheric conditions which cannot be estimated from the surges with our simple LLR model, in particular wind directions and intensity." This seems unlikely given the periods of time of several days and the fact that the data are smoothed. I wonder how these comparisons are with a simple inverted barometer approach, not using the LLR model.

---

## Author Response (AR2)

**Answer to 2nd round of review of manuscript:**
**"Could old tide gauges help estimate past atmospheric variability?"**
**for *Climate of the Past**

Paul Platzer, Pierre Tandeo, Pierre Ailliot, Bertrand Chapron

August 2024

*Reviews in black,* *answers in blue,* ***quotes from revised manuscript in bold****.*

The authors have made substantial changes to the manuscript in response to my previous concerns. I still believe that this idea is original and worth exploring. Again, I am not commenting on the statistical model, as it is beyond my expertise. The results though seem to make sense at constraining the ensemble members in 20CR dataset. Despite the undoubtful interest of the work, I still have a number of comments on the manuscript, some minor, some other that would require further exploration. I am listing all of them below, as they appear in the text. Many of my comments arise from wrong statements or parts of the text that are unclear. This makes the manuscript sometimes hard to follow, even in the descriptive parts.

My first comment is about the format of the author tracked changes document. Please, avoid this practice in future submissions. It makes impossible for the reviewer to go through this file and identify the changes and it makes reading very uncomfortable. This is not what I understand should be a version with tracked changes which should contain a comparison between the original and the revised versions. I have therefore gone through the new file, without paying much attention to the tracked version. All the lines below thus refer to the new version.

We thank the reviewer for the new comments and suggestions which again helped make our work much more precise and rigourous.

We have used another tool to create author track-changes, and we hope it allows for a more comfortable reading. In this response document, the underlined line numbers refer to the 2nd revision of the manuscript, but we also include the line numbering of the tracked-changes document.

Comment-by-comment responses are listed below.

**1.** The term surge is misleading. Please, use the terminology as defined in the literature to avoid misunderstandings: https://link.springer.com/article/10.1007/s10712-019-09525-z.

**1.** We thank the reviewer for this clarification. The ambiguous term "surge" was replaced by "surge residual" in most cases, and we have also clarified the additional operations that we make (removing the yearly median, making averages, etc.) where needed. See the response to other comments below.

**2.** Abstract: the storm surge is not the response to atmospheric pressure. It also includes winds and waves and mean sea level.

**2.** Indeed. We have stressed this out by using the word "*including*", writing
"The surge residual is the non-tidal component of coastal sea-level. It responds to the atmospheric circulation, including the direct effect of atmospheric pressure on the sea-surface."

**3.** l. 72: .3 mb should be 0.3 mbar (I guess)

**3.** Corrected.

**4.** l. 93 "As the latter is driven by a physical phenomenon called the "inverse barometer effect". This statement is incorrect, see my comment and reference above.

**4.** This was changed to, l. 93 (also 93 in tracked-changes document):
"As the latter is driven, in part, by a physical phenomenon called the 'inverse barometer effect' "

**5.** l. 92-97. This is probably a good approximation but it is not strictly correct. First, the anomaly of every ensemble member should be calculated with its own mean. Second, the average should be calculated within the global ocean. According to the responses to my previous revision, the latter has been checked and made negligible differences. You should need to check the first point though.

**5.** To check the first point, we have computed :

1. The world-ocean average for each individual member, in year 1870. Then, at each time-step we computed the standard deviation of this average over all 80 members. This first quantity measures intra-member variability in ocean average pressure.

2. The standard deviation over all 80 members of the MSLP at the city of Brest. This second quantity measures the variability of the intra-member variability in local pressure.

We have checked that the first quantity is almost steady, between 22Pa and 35Pa. The second quantity displays more variability, and varies between 200Pa and 800Pa. It thus appears that the potential contribution of the inter-member variability in ocean-averaged pressure to the results of our study is weak.

Furthermore, our estimates of pressure based on the LLR shown in section 5 necessary rely on a unique estimate of the ocean average pressure, for which we have used the average over the 20CRv3 members. It thus makes sense to use this members-average throughout the article instead of one ocean-average per member, although the second solution would be more accurate.

Therefore, we have chosen to keep the computations as is, but to mention this approximation with the following two sentences l. 98 (also l. 98 in tracked-changes document) :

"Note that there is a small variability in ocean-averaged pressure between 20CRv3 members in the 19th century. However, we have checked that this variability is one order of magnitude smaller than the inter-member variability of MSLP at the city of Brest, which justifies our approximation of using simply the members-average of the ocean-averaged pressure as a reference."

**6.** please remove figure 2. It does not make much sense to have only one station. This can be mentioned in the text.

**6.** The figure was removed.

**7.** l. 112: storm surge and skew surge are two different metrics

**7.** This was removed.

**8.** l. 112: "To access the surge, one first has to remove the tidal part of the signal, and then to remove yearly variations of the mean-sea-level (at interannual anddecadal scale), such as sea-level rise". This is incorrect. The storm surge is the difference between sea level and tides, so it does include mean sea level, so the definition is not exact. Then mean sea level can be removed if the purpose is to understand short term changes.

**8.** We thank the reviewer for this clarification. We have modified the definitions and added the reference suggested above, giving l. 114 (l. 116 in tracked-changes document):
"

As mentioned earlier, the part of the sea-level which responds to atmospheric processes is the surge residual (see definition in Gregory et al., 2019). To access the surge residual, one has to remove the tidal part of the signal. Then, as we are interested in sub-seasonal variations, we also remove the yearly variations of the mean-sea-level (at interannual and decadal scale), such as sea-level rise (Cazenave and Llovel, 2010). In this work, we also use moving averages and differences of the surge residual. All these steps are exemplified in Fig. 2.

We first compute the tidal constituents of the raw sea-level (blue curve, Fig. 2.a) using U-Tide (Codiga, 2011), which performs harmonic (Fourier) decomposition with prescribed frequencies corresponding to planetary movements. The tidal constituents are computed over two different periods, one is 1847-1890, and the second is 1981-2015. Removing the tidal part of the signal gives the surge residual (orange dashed line of Fig. 2.a), which has a temporal average value of ∼4m for the Brest tide gauge.

Then, we remove the yearly median value of the sea-level (orange dashed line of Fig. 2.b). We choose to remove the median and not the mean because the mean can in principle be influenced by the number and magnitude of extremes in a given year, which can be linked to the number and magnitude of storms passing in a given year. This second step allows to access the zero-median surge residual which is noted $h(t)$ in the following:

$$h(t) = H(t) - \text{Tide}_H(t) - \text{median}\left[H(t'),\, t' \in \text{year}(t)\right] \,, \tag{1}$$

where $H(t)$ denotes the raw sea-level, $\text{Tide}_H(t)$ is the tidal part of the signal computed from $H$, and $\text{year}(t)$ is the year in which time $t$ is found.
"

**9.** figure 3: in the legend and caption, level should be sea level

**9.** This was modified (now in Figure 2).

**10.** l. 127: "These oscillations are either due to tide-surge interactions (Horsburgh and Wilson, 2007) or to measurement errors in the 19th century leading to phase shifts" So you mean that these are 12-h oscillations? it is hard to see in the figure.

**10.** First, when performing 12h-averages we filter *more* than 12h-oscillations, and as pointed by the reviewer we sometimes see oscillations at smaller frequencies that are removed as well. Therefore we have rephrased to give l. 131 (l. 134 in tracked-changes document):

"Note from Fig. 2.b that the surge residual fluctuates at hourly scale, part of which are oscillations which are not due to variations in atmospheric pressure. For instance, these oscillations can be due to tide-surge interactions (Horsburgh and Wilson, 2007) or to measurement errors in the 19th century leading to phase shifts."

Also, we have chosen another example where 12h-oscillations are more obvious. This is also highlighted in the text with the following sentence l. 133 (l. 136 in tracked-changes document):

"Such 12h-oscillations can dominate the surge residual signal in Brest where the tidal amplitude is large (see for instance on the 29th and 30th of January 2014, Fig. 2)."

**11.** l. 131: "This also implies that these 12-hours-averaged surges will only respond to atmospheric events persisting for more than 12 hours." I do not think this is correct. It will smooth higher frequency changes but not remove them completely.

**11.** We chose to simply remove this sentence.

**12.** l. 135-142: the inverted barometer acts at periods shorter than 12h, so the statement about longer periods is incorrect. Also, what do you mean by "local time-variations"? In any case, because this record is to be compared to a very low resolution model output, this approach may not be critical. I don't think it is applicable to high resolution models though.

**12.** We have reformulated the rationale for including the variable $\Delta \overline{h}^{3h}(t) := \frac{1}{3} \sum_{t'=-2}^{t'=+1} [h(t+t') - h(t-12+t')]$ in our LLR. It is now as follows l. 141 (l. 145 in tracked-changes document):

" Finally, note that if atmospheric pressure variations are faster than the typical time of adjustment of sea-level, one expects deviations from the inverse barometer approximation (Bertin, 2016). Therefore, fast time-variations of the surge residual are also expected, statistically speaking, to be associated with deviations from the inverse barometer approximation. To allow the model described in section 3.1 to capture this effect, we compute the difference between the surge at time $t$ and at time $t - 12h$, choosing the 12h-interval again to filter out oscillations at a period close to 12h. Furthermore, since the reanalysis is run at 3h-resolution, we perform a 3h-moving average of the surge residual before computing the difference. This difference is noted $\Delta \overline{h}^{3h}(t)$ and defined by the following equation: "

**13.** l. 151-156: this is not clear to me. I understand that pressure observations have biases. Are you correcting for yearly dependent biases? If so, this is altering the temporal variability of the observations. This means that they cannot be compared as independent from the model. Bias-corrections must apply the same bias during the entire record. If the record is made of two separate set of observations then it makes sense to apply two biases separately, but never every year.

**13.** We have modified our procedure, applying one bias-correction for each dataset, as described l. 156 (l. 163 in tracked-changes document):

" We have found a shift in average pressure between the EMULATE and Météo France datasets. To overcome this issue, and since we are only interested in sub-seasonal atmospheric variability, we added a constant value of $\sim 0.22$hPa for the period 1860-1880 (EMULATE dataset) to each value of the independent pressure observation datasets, so that the average pressure are equal between the independent observed pressure and the 20CRv3 mean pressure linearly interpolated at the city of Brest. We did the same operation for the period covered by the Météo France dataset that we are using (1855-1859 and 1881-1894), adding a value of $\sim 7.18$hPa. "

We have reproduced the figures using this new bias correction (originally numbered Fig. 9, 11, 12, 13, 14, now numbered 8, 10, 11, 12, 13) and found no noticeable difference. Therefore, we kept the figures as is.

**14.** l. 169-171: "Statistical correlation between pressure variations and wind intensity and direction are responsible for deviations from the inverse barometer approximation of the statistical linear relationship between surges and pressure (Ponte, 1994)" I am unsure about what this means. What statistical correlation? And after this: "As a consequence of these combined effects of wind and pressure, the statistical relationship between the filtered surges and the pressures from 20CRv3 is expected to be non-linear, and not deterministic. " Again, unclear. I would say that the effect is linear but there are other processes. Next: "As showed by Hawkins et al. (2023), using a physical coastal model forced by the values of pressure (and winds) from the 20CR can lead to biases in the estimation of associated surges due to the resolution of the reanalysis, so that a statistical model is needed to correctly represent uncertainties" The need of a statistical model is not really an implication of the inability of the coarse resolution numerical models.

**14.** We have reformulated this paragraph, hoping that it is now more accurate and clear l. 177 (l. 187 in tracked-changes document):

" Since wind is not included in our model, the relationship between the filtered surge residuals and the atmospheric pressures from 20CRv3 should not be deterministic. Also, it is likely that typical wind conditions depend on the amplitude of MSLP anomaly, so that the average value of MSLP anomaly for a given value of surge residual

[Figure]

Figure 1: Reproducing a figure of the manuscript but with a simple inverted barometer instead of the more complex LLR.

in Brest may be a non-linear function. As showed by Hawkins et al. (2023), using a physical coastal model forced by the values of pressure (and winds) from the 20CR can lead to biases in the estimation of associated surges due to the resolution of the reanalysis. A statistical model can thus be used as a tool to correct such biases and represent uncertainties. In our case, since we want to estimate pressure based on the surge residuals only, the effect of unknown wind or other processes must also be taken into account through uncertainty quantification. "

**15.** L. 211-215 on the validation of the LLR model. If I understood correctly, the model is applied to a part of the dataset for the period 1980-2015 (those for which neighbours are 14 days apart). Then, Figure 4 should represent the comparison between the true and the modelled values only for those part of the time series that were not used to fit the model. Is this what it is showing? From the caption and text it seems that it is showing all values, but in this case, it is not a validation because values used to fit the model are also in there. Please, clarify.

**15.** In our procedure, for each time $t \in [1980 - 2015]$, we use 200 samples in $[1980 - 2015]$ but excluding $[t - 14$ days , $t + 14$ days] to fit a linear regression. Therefore, the data that is used to fit the model does not include the true values. The 200 samples are chosen at each time $t$ using a similarity criterion (smallest Euclidean distance) on the vector $\left[ \overline{h}^{12h}(t), \Delta \overline{h}^{3h}(t) \right]$.

When using the LLR on the 19th century data, since we search for neighbours in the period $[1980 - 2015]$, we do not need to use the twice-14-days window exclusion.

We have reformulated, l. 220 (l. 231 in tracked-changes document):

" To test the accuracy of this model on the 1980-2015 period, we apply it for all times $t \in [1980 - 2015]$, searching for neighbours' times $t_i$ in the same period but with the condition that there is a minimum of two weeks between $t$ and $t_i$ (i.e., excluding the interval $[t - 14$ days , $t + 14$ days]). This is called the 'leave-one-out' procedure, ensuring that the data that is used to fit the model does not include the true values. "

**16.** How are the results of the LRR different from purely inverted barometer approximation? Is Figure 4 improved with the LRR with respect to the most simple approach? In principle, if only subseasonal pressure variations are targeted, inverted barometer is likely a good proxy for sea level changes. If another more complex model is to be used, its benefits should be demonstrated. I am afraid that at this point it is still unclear to me why this model is needed.

**16.** For comparison, we show here in Fig. 1 the same plots as in Figure 4 of the 1st-revision of the manuscript but for a simple inverted barometer, understood here as a linear regression between 1. the difference between the local MSLP and ocean-averaged MSLP and 2. the sea-level residual as defined in the text of the 1st-revision of the manuscript.

The left panel of this figure (Fig. 1.a) shows no clear sign of improvement in average prediction with the LLR model compared to a simple IB. However, the right panel (Fig. 1.b) shows that the rescaled IB differs more from the standard normal distribution than the LLR (higher peak probability around zero), indicating that the LLR provides better uncertainty quantification. This is because the variance "var$(t)$" as defined in the revised manuscript is situation-dependent. A simple linear regression assumes *homoscedasticity* (i.e., that the distribution of the difference between the linear prediction and the true value does not depend on the value itself). On the contrary, the LLR assumes *heteroscedasticity*. This allows for better uncertainty quantification and is a key advantage of the LLR. As we are interested in uncertainty quantification, this justifies the use of a more complex algorithm such as the LLR.

More details on the interest of the LLR are given in the answer to question **18.** in this response document.

**17.** l. 358-359: "In 1865 (Fig. 9.b), although the surge-based reconstruction happens to be more consistent with observations than the reanalysis, the reverse is also true." Please, rephrase, two opposite things cannot be true ...

**17.** This was poorly formulated and brought confusion. We have reformulated to give l. 368 (l. 381 in tracked-changes document):

" In 1865 (Fig. 8.b), although the surge residual-based reconstruction is sometimes more consistent with observations than the reanalysis, there are as many occasions where it is the reanalysis which is more consistent with the independent observations. "

**18.** l. 362: "We attribute these biases to different atmospheric conditions which cannot be estimated from the surges with our simple LLR model, in particular wind directions and intensity." This seems unlikely given the periods of time of several days and the fact that the data are smoothed. I wonder how these comparisons are with a simple inverted barometer approach, not using the LLR model.

**18.** For comparison purposes and at the demand of the reviewer, we reproduce here Fig. 8 of the second revision of the manuscript (numbered Fig. 9 in the first revision of the manuscript), but replacing the results of the LLR with the results of a simple linear regression between surge residuals and atmospheric pressure residuals, simply referred to as "IB" in the figure's legend (Fig. 2 in this response document). A first observation is that the uncertainties associated with the simple IB approach are larger. To allow for a more in-depth comparison, we also plot all estimates in one figure (see Fig. 3 of this response document). We see that the LLR uncertainty range is always comprised in the IB uncertainty range, although smaller. The average from the LLR is close to the IB average. At times, some of which are highlighted in green, the LLR is closer to the independent observations (and the 20CRv3 reanalysis) than the simple IB. Note that this is not always true, and there are a few counter-examples. However, the reduced uncertainties show that the LLR is overall slightly more precise than a simple IB.

**References**

Xavier Bertin. Storm surges and coastal flooding: status and challenges. *La Houille Blanche*, (2):64–70, 2016.

Anny Cazenave and William Llovel. Contemporary sea level rise. *Annual review of marine science*, 2:145–173, 2010.

Daniel L Codiga. Unified tidal analysis and prediction using the utide matlab functions. 2011.

Jonathan M Gregory, Stephen M Griffies, Chris W Hughes, Jason A Lowe, John A Church, Ichiro Fukimori, Natalya Gomez, Robert E Kopp, Felix Landerer, Gonéri Le Cozannet, et al. Concepts and terminology for sea level: Mean, variability and change, both local and global. *Surveys in Geophysics*, 40:1251–1289, 2019.

Ed Hawkins, Philip Brohan, Samantha N Burgess, Stephen Burt, Gilbert P Compo, Suzanne L Gray, Ivan D Haigh, Hans Hersbach, Kiki Kuijjer, Oscar Martínez-Alvarado, et al. Rescuing historical weather observations improves quantification of severe windstorm risks. *Natural hazards and earth system sciences*, 23(4):1465–1482, 2023.

KJ Horsburgh and C Wilson. Tide-surge interaction and its role in the distribution of surge residuals in the north sea. *Journal of Geophysical Research: Oceans*, 112(C8), 2007.

[Figure]

Figure 2: Reproducing a figure from the article but replacing the LLR with a simple corrected inverted barometer (linear regression between surge residuals and pressure anomaly, simply referred to as IB in the legend).

[Figure]

Figure 3: Same as previous figure, but showing all estimations at once. IB uncertainty ranges are in black, and LLR uncertainty ranges are in orange.